# LAPO: Latent-Variable Advantage-Weighted Policy Optimization for Offline Reinforcement Learning

**Xi Chen[1], Ali Ghadirzadeh[2], Tianhe Yu[2], Jianhao Wang[1], Yuan Gao[3],**
**Wenzhe Li[1], Bin Liang[1], Chelsea Finn[2], Chongjie Zhang[1]**
[1]Tsinghua University, China, [2]Stanford University, USA
[3]Shenzhen Institute of Artificial Intelligence and Robotics for Society, China

## Abstract

Offline reinforcement learning methods hold the promise of learning policies from pre-collected datasets without the need to query the environment for new samples. This setting is particularly well-suited for continuous control robotic applications for which online data collection based on trial-and-error is costly and potentially unsafe. In practice, offline datasets are often *heterogeneous*, i.e., collected in a variety of scenarios, such as data from several human demonstrators or from policies that act with different purposes. Unfortunately, such datasets often contain action distributions with multiple modes and, in some cases, lack a sufficient number of high-reward trajectories, which render offline policy training inefficient. To address this challenge, we propose to leverage latent-variable generative model to represent high-advantage state-action pairs leading to better adherence to data distributions that contributes to solving the task, while maximizing reward via a policy over the latent variable. As we empirically show on a range of simulated locomotion, navigation, and manipulation tasks, our method referred to as latent-variable advantage-weighted policy optimization (LAPO), improves the average performance of the next best-performing offline reinforcement learning methods by 49% on heterogeneous datasets, and by 8% on datasets with narrow and biased distributions.

## 1   Introduction

Offline reinforcement learning (RL), also known as batch RL [26], addresses the problem of learning an effective policy from a static fixed-sized dataset without interacting with the environment to collect new data. This formulation is especially important for robotics, as it avoids costly and unsafe trial-and-error and provides an alternative way of leveraging a pre-collected dataset. However, in practical settings, such offline datasets are often heterogeneous and are collected using different policies, leading to a data distribution with multiple modes. These data-collection policies may aim to accomplish tasks that are not necessarily aligned with the target task or may accomplish the same task but provide conflicting solutions. In contrast to the prior works that have focused on the distributional shift problem [24, 9, 25], in this paper, we address the problem of learning from heterogeneous data.

The main challenges in learning from heterogeneous settings are the existence of conflicting actions in the dataset, and the lack of sufficient high-return trajectories as the dataset is constructed by re-labeling state-action pairs from some different tasks. Learning from heterogeneous datasets with conflicting actions is particularly challenging for implicit policy constraint methods that formulate a supervised learning objective function based on the forward Kullback–Leibler (KL)-divergence between the parametric policy being learned and the closed-form optimal policy found through advantage-weighted behavior cloning [39, 41, 44]. Figure 1 illustrates an example of offline heterogeneous settings in which state-conditional distributions learned directly over the action space

36th Conference on Neural Information Processing Systems (NeurIPS 2022).

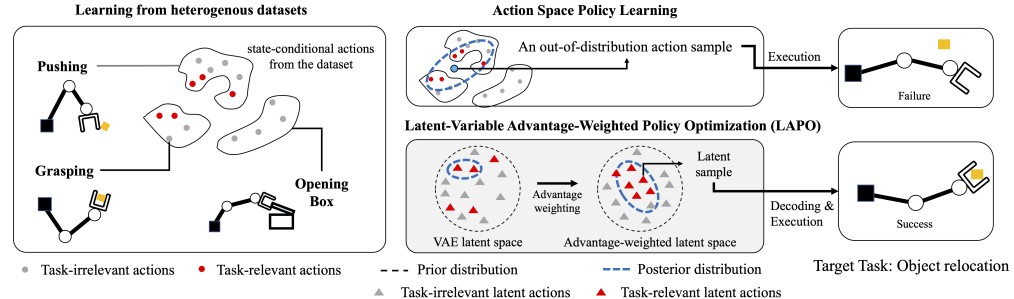

Figure 1: An example scenario of learning from data with heterogeneous action distributions. (left) The dataset includes data from three different tasks, pushing, grasping, and opening boxes. (right) The learned task is to relocate an object to a goal position. (middle up) Learning policies directly in the action space can result in sampling from out-of-distribution actions which subsequently can fail the learning task. (middle bottom) LAPO constructs a state-conditional latent space and learns a latent policy to only select in distribution high-return actions.

using the forward KL-divergence objective may assign high probability to out-of-distribution (OOD) actions which results in sub-optimal policies. Replacing the forward-KL-divergence with a reverse KL-divergence can avoid this issue in theory, as it forces the policy to only learn a single mode of the distribution. But in practice, it requires querying the behavior policy which is unknown, and using an erroneous approximation of the behavior policy can negatively affect the performance ([39]). A similar problem has been reported when explicitly constraining the policy on multi-modal distributions using maximum mean discrepancy (MMD) distances [57]. This problem however may seem to be solved by using more expressive policies such as variational auto-encoders (VAEs) or Gaussian mixture models (GMMs). However, in practice this does not help in heterogeneous settings with a low density of high-return trajectories. In such settings, adhering to the data distribution using policy constraint methods can result in sub-optimal performances.

We illustrate this problem with a toy navigation task in which an agent navigates to a goal position while avoiding an obstacle. In this example, the demonstrated expert actions have two prominent modes at the initial state corresponding to moving to either the right or left side of the obstacle. We assume moving to the right leads to higher returns, but we have a lower density of such actions in the dataset. In Figure 2, we show the action distribution learned by different offline RL algorithms, and in Table 1, we present their averaged performance. Figure 2.a illustrates the histogram of actions sampled from a trained policy with AWAC [39], a policy constraint method. The method fails in this setting since the learned action distribution includes OOD actions which result in a collision with the obstacle. Learning a GMM policy with AWAC can avoid the OOD action problem but still assigns a high probability distribution over low-return actions (Figure 2.b). Other offline RL methods, including BCQ [9] and PLAS [57], which learn a VAE policy, also fail to represent the high-return actions in this setting. PLAS can only model the low-return but high-density samples in the dataset (Figure 2.c).

Figure 1 also illustrates a potential solution to the problem of learning from heterogeneous datasets. The intuition is to construct a state-conditioned latent space that represents high-return actions. In this case, we can learn simple state-conditional distributions such as Gaussian distributions over the latent space which, as shown in Figure 2.d, can capture task-relevant and high-reward actions without including out-of-distribution samples. Based on this intuition, we propose to learn a latent-space policy by alternating between training the policy and maximizing the advantage-weighted log-likelihood of data. This biases the RL policy to choose actions supported by the training data while effectively solving the target task.

The main contributions of this work is the introduction of a new method, which we refer to it as latent-variable advantage-weighted policy optimization (LAPO), that can efficiently solve heterogeneous offline RL tasks. LAPO learns an advantage function and a state-conditional latent space in which high-advantage action samples are generated by sampling from a prior distribution over the latent space. Furthermore, following a prior work, [57], we also train a latent policy that obtains state-conditioned latent values which result in higher reward outcomes compared to latent samples directly

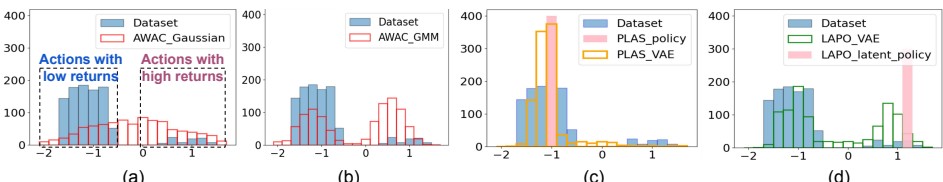

Figure 2: The histogram of actions for the toy navigation task at the initial state in the offline dataset (in blue), and the histogram of actions sampled from (a) the AWAC Gaussian policy (in red), (b) the AWAC GMM policy (in red), (c) the PLAS generative model (in yellow) and the PLAS latent policy (in pink), and (d) the LAPO generative model (in green), and the LAPO latent policy (in pink).

Table 1: The performance of policies in the toy navigation task learned by AWAC, PLAS and LAPO. The return is averaged by $1,000$ samples. The agent receives a reward of +1 to move left, $a \in [-1.5, -0.5]$, +10 to move right, $a \in [0.5, 1.5]$, and 0 otherwise.

| | AWAC | | PLAS | | LAPO | |
|---|---|---|---|---|---|---|
| | Gaussian | GMM | VAE | policy | VAE | latent_policy |
| Average return | 2.43 | 4.39 | 1.03 | 1.00 | 4.24 | **10.00** |

drawn from a prior distribution. We compare LAPO to vanilla behavior cloning, BCQ [9], PLAS [57], AWAC [39], IQL [23], and CQL [25] on a variety of simulated locomotion, navigation, and manipulation tasks, provided heterogeneous offline datasets and also biased datasets with narrow data distribution in standard offline RL benchmarks. LAPO is the only method that yields good performance across all of these tasks and on average improves by 49% over the next best method for heterogeneous datasets, and by 8% on other offline RL tasks with narrow data distributions.

## 2 Preliminaries

The goal of reinforcement learning is to obtain a policy that maximizes a notion of accumulated reward for a task as a Markov decision process (MDP). In offline RL settings, we assume that we are given a dataset of tuples $\mathcal{D} = \{(s_t, a_t, r_t, s_{t+1})\}$, where, $s_t \in \mathcal{S}$ and $a_t \in \mathcal{A}$ denote the state and action at the time step $t$, $r_t = r(s_t, a_t)$ is the reward given by a bounded reward function, and $s_{t+1}$ is sampled from a fixed transition probability distribution $p(s_{t+1}|s_t, a_t)$. Note that actions can come from a mixture of policies which is referred to as the behavior policy. The policies may try to accomplish different tasks and hence generate trajectories unrelated to the target task.

Given an initial state distribution $\mu_0(s)$ and a discount factor $\gamma \in (0, 1)$, the RL agent optimizes a policy $\pi(a|s)$ to maximize the expected cumulative reward

$$J(\pi) = \mathbb{E}_{s_0 \sim \mu_0, s_{t+1} \sim p(\cdot|s_t, \pi(s_t))} \left[ \sum_{t=0}^{\infty} \gamma^t r(s_t, \pi(s_t)) \right]. \tag{1}$$

The state action value function (Q function), of the policy is defined as $Q^\pi(s, a) = \mathbb{E}_\pi[\sum_{t=0}^{\infty} \gamma^t r_t | s_0 = s, a_0 = a]$, which can be approximated with the Bellman equation:

$$\phi^* = \arg\min_{\phi} \mathbb{E}_{(s_t, a_t, r_t, s_{t+1}) \sim \mathcal{D}} \left[ r_t + \gamma V(s_{t+1}) - Q_\phi(s_t, a_t))^2 \right], \tag{2}$$

where, $\phi$ denotes the parameters of the Q-function, $V$ denotes the value function $V(s) = \mathbb{E}_{\pi(a|s)}[Q_{\phi'}(s, a)]$ and can be approximated by sampling actions from the policy distribution and averaging the corresponding Q value. $\phi'$ denotes the parameters of the Q-function at the previous iteration. The advantage of a pair of state and action is then defined as $A(s, a) = Q_\phi(s, a) - V(s)$.

## 3 Implicit Policy Constraints with KL-divergence

A central challenge for offline RL methods is to limit the state-conditional action distribution of the learned policy to the empirical conditional action distribution of the dataset. It is very likely for off-policy algorithms to have overly optimistic values, as they tend to query the Q-function for action

inputs outside the training distribution. Policy constraints methods [24, 51] deal with this problem by keeping the learned policy close to the behavior policy using the following constrained policy optimization formulation:

$$\operatorname*{argmax}_{\pi} \mathbb{E}_{s\sim\mathcal{D}} \mathbb{E}_{\pi(a|s)}[A(s,a)], \text{s.t.} \ \mathbb{E}_{s\sim\mathcal{D}}[D(\pi_\theta(a|s), \pi_\beta(a|s))] < \epsilon \qquad (3)$$

where, $\pi_\beta(a|s)$ denotes the unknown empirical conditional action distribution of the dataset, and $D$ denotes a distance measure such as KL-divergence [39, 44] or maximum mean discrepancy (MMD) [24], and $\epsilon$ is a threshold parameter. In case, we use the KL-divergence as the divergence in Equation 3, the optimal $\pi^*$ can be expressed as

$$\pi^*(a|s) \propto \pi_\beta(a|s) \exp(A(s,a)/\lambda), \qquad (4)$$

where, $\lambda$ is a temperature parameter that depends on the $\epsilon$.

Prior work [39, 44, 41] suggested to incrementally solve Equation 4 by representing the optimal policy $\pi^*(a|s)$ as a non-parametric policy, and then project it onto the parametric policy $\pi_\theta(a|s)$ via supervised regression:

$$\operatorname*{argmin}_{\theta} \mathbb{E}_{\mathcal{D}}[\mathrm{D}_{\mathrm{KL}}(\pi^*(a|s)||\pi_\theta(a|s))] = \operatorname*{argmax}_{\theta} \mathbb{E}_{\mathcal{D}}[\mathbb{E}_{\pi^*(a|s)}[\log \pi_\theta(a|s)]], \qquad (5)$$

where, the expectation is estimated by sampling from the dataset. In practice, the non-parametric policy can be implemented by weighting samples from the dataset $\mathcal{D}$ using importance weights $\omega = \exp(A(s,a)/\lambda)$, and the projection via KL-divergence can be done via weighted supervised learning using these weights [29].

# 4 Latent-Variable Advantage-Weighted Policy Optimization

In this work, we aim to address the problem of learning from heterogeneous offline datasets collected by policies with different purposes. The challenge arises from the fact that the dataset contains conflicting actions distributed with multiple modes. Besides, the dataset often does not include a sufficient number of high-return trajectories to learn the task properly, as the dataset may be constructed by re-labeling rewards for state-action pairs collected for some other tasks that not necessarily aligned with the target task. We argue that in this setting we not only need a more expressive policy class to represent the multi-modal action distributions, but also need a mechanism that can select which samples to use for the given target task.

We present a novel method, named *latent-space advantage-weighted policy training* (LAPO), which learns a policy by maximizing the Q function over a latent space that is trained to approach the distribution of actions that leads to higher returns in solving the task. The overall objective of the LAPO can be defined as follows:

$$\operatorname*{argmax}_{\vartheta,\theta} \mathbb{E}_{\substack{z\sim\pi_\vartheta(\cdot|s)\\a\sim p_{\theta'}(\cdot|s,z)\\s\sim\mathcal{D}}} [Q_\phi(s,a)] + \mathbb{E}_{(s,a)\sim\mathcal{D}}[\omega * \log[p_\theta(a|s)]]. \qquad (6)$$

The objective consists of two main components: (1) an advantage-weighted log-likelihood of the actions in the dataset and, (2) a state-action value function. This objective is optimized iteratively by alternating between maximizing each component separately. The weighted log-likelihood is maximized by training a latent-variable state-conditioned generative model $p_\theta(a|s) = \mathbb{E}_{z\sim p(z)}[p_\theta(a|s,z)]$. Here, $z$ denotes a latent variable, $p(z)$ is the prior distribution, $p_\theta(a|s,z)$ is a generative model (decoder), and $\theta'$ denotes the parameters of the decoder in the previous iteration. This component extends advantage-weighted regression (AWR, Equation 5) [41] using more expressive policy classes. Also, as illustrated in Figure 2c and d, weighting by advantages allows the generative model to effectively represent high-return state-action transitions. The state-action value function $Q_\phi(s,a)$ is updated in every iteration, and is then improved by updating the latent policy $\pi_\vartheta(z|s)$. The latent policy outputs a latent variable $z$ which is then mapped to an action $a \in \mathcal{A}$ by the decoder $p'_\theta(a|s,z)$ with the parameters updated in the previous iteration. The latent policy is trained with a standard actor-critical RL approach to maximize the state-action value function. Next, we will describe each of these components in detail.

**Maximizing the weighted log-likelihood:** LAPO incrementally learns to maximize the weighted log-likelihood by alternating between two steps: (1) estimating the importance weight of each action,

---

**Algorithm 1** Latent-Variable Advantage-Weighted Policy Optimization

---

**Input:** $\mathcal{D} = \{(s_t, a_t, s_{t+1})_i\}, i = 1, \ldots, N$.
**Initialize:** $\phi, \theta, \psi, \vartheta$ randomly, $\theta' = \theta$, and $\omega = 1.0$.
**repeat**
    Update the decoder $p_\theta$ using Equation 7.
    Update the Q-function $Q_\phi$ using Equations 2 and 8.
    Estimate the advantage weights for every state-action pair.
    Update the latent policy $\pi_\vartheta$ using Equation 9
    Update $\theta'$ according to $\theta$
**until** $M$ iterations completed

---

and (2) regressing the actions in the dataset with the importance weight of each action. In practice, to characterize the importance of high-return trajectories, we can use the exponential advantages as the importance weight, $\omega = \exp(A(s, a)/\lambda)$, where $\lambda$ is a temperature parameter. Following [21], we derive a weighted variational lower bound that maximizes the weighted log-likelihood of data:

$$\max_{\theta,\psi} \mathbb{E}_{s,a\sim\mathcal{D}}[\omega \, \mathbb{E}_{q_\psi(z|s,a)}[\log(p_\theta(a|s,z)) \, - \beta \, \mathrm{D_{KL}}(q_\psi(z|s,a) \, || \, p(z))\,]], \tag{7}$$

where $q_\psi(z|s,a)$ is the variational posterior distribution, $p(z)$ is the prior distribution over the latent variable and typically modelled by a standard normal distribution, and $\beta$ is a hyper-parameter to balance the two loss terms [17].

**Policy evaluation:** As explained earlier, the estimated advantage of each state-action pair is used to update the decoder. These advantage estimations are incrementally found through a policy evaluation process in which the overall policy, formed by combining the latent policy and the decoder, is being evaluated. The state-conditional action distribution of the overall policy $\pi_{\vartheta,\theta'}(a|s)$, is found by first sampling a latent value from the latent policy $z \sim \pi_\vartheta(z|s)$, and then feeding the latent value to the decoder to sample the action $a \sim p_{\theta'}(a|s,z)$. In the policy evaluation step, we update the value function by minimizing the squared temporal difference error using Equation 2. To calculate the expected value of a state, we sample actions from the overall policy, i.e.,

$$V(s) = \mathbb{E}_{a\sim p_{\theta'}(\cdot|s,z),z\sim\pi_\vartheta(\cdot|s)}[Q_{\phi'}(s,a)]. \tag{8}$$

Since the latent values $z$ will be used as the input to the decoder, an unbounded $z$ can lead to generating out-of-distribution actions. Therefore, to better ensure sampling in-distribution actions, $\pi_\vartheta(z|s)$ can be modeled as a truncated Gaussian [5]. For deterministic versions of the latent policy, the policy output can be limited to $[-z_{\max}, z_{\max}]$ using a $\mathrm{tanh}$ function.

**Policy improvement:** LAPO optimizes the latent policy $\pi_\vartheta(z|s)$ in every iteration to directly maximize the return. It is trained based on standard RL approaches such as DDPG [30] or TD3 [8] to maximize the RL objective in Equation 1, by learning latent actions which result in high returns after being converted to the original action space using the decoder.

The latent policy is updated by maximizing the Q-function over states sampled from the dataset, latent variables from the latent policy, and actions from the decoder $p_{\theta'}$:

$$\underset{\pi_\vartheta}{\mathrm{argmax}} \, \mathbb{E}_{a\sim p_{\theta'}(\cdot|s,z),z\sim\pi_\vartheta(\cdot|s),s\sim\mathcal{D}}[Q_\phi(a,s)]. \tag{9}$$

A summary of our method is presented in Algorithm 1. We randomly initialize the parameters of the state-action Q function, the decoder, the amortized variational distribution, and the latent policy. In every training iteration, we first update the decoder and the variational distribution $q$ using Equation 7 given the latest values of the advantage weights. Then, we update the Q function using Equations 2 and 8 provided the updated action and latent policies. Then, for every state-action pair in the dataset, we compute the advantage using the updated Q function, the latent policy and the decoder, and then estimate the weights. Finally, we update the latent policy using Equation 9.

## 5 Related Works

**Offline Reinforcement Learning:** Offline RL methods generally address the problem of distribution shift between the behavior policy and the policy being learned [9, 24], which can cause issues

due to out-of-distribution actions sampled from the learned policy and passed into the learned critic. To address this issue, prior methods constrain the learned policy to stay close to the behavior policy via explicit policy regularization [32, 18, 51, 24, 22, 13], via implicit policy constraints [44, 49, 42, 41, 39, 22, 23, 53, 35], by regularizing based on importance sampling [37, 31, 50, 52], by learning of conservative value functions [25, 46], by leveraging auxiliary behavioral cloning losses [7, 38], and through model-based training with conservative penalties [56, 20, 3, 47, 36, 28, 55]. Compared to these prior works, we opt to develop a method that uses an implicit policy constraint. However, prior implicit policy constraint methods have been largely limited to Gaussian policies, leading to OOD action samplings and sub-optimal performance on heterogeneous datasets, We overcome this challenge by introducing a new method that leverages latent variable models. As we will find in Section 6, our method also outperforms state-of-the-art prior methods in other categories, particularly when learning from heterogeneous datasets.

Prior works [1, 19, 54] also studied the problem of learning from heterogeneous dataset in the offline RL with heterogeneous dynamics setting [1] and multi-task offline RL with data sharing scenario [19, 54] respectively. Both settings require knowledge of the agent/task identifier. Here, we study a more general problem setting including datasets with conflicting actions without assuming access to the ground-truth identifier of the source of the heterogeneity.

**RL with Generative Models:** Generative models have been used by prior works for improving training performance [45, 11, 12], enabling transfer learning [4, 10], implementing hierarchical RL [2, 33, 34, 40], avoiding distributional shift in offline RL settings [9, 57], and learning dynamics models [27, 16, 43, 15]. Our proposed method resembles PLAS [57], which learns a generative model and latent policy; but, unlike this prior work, our method trains an advantage-weighted generative model by alternating between learning the generative model, the advantage function, and the latent policy. This allows LAPO to capture different high-reward solutions using a simple Gaussian distribution in the latent space. As we will find in Section 6, this distinction is crucial, as LAPO significantly outperforms PLAS on heterogeneous datasets.

# 6  Experiments

Our experiments aim to answer the following questions: (1) How does LAPO compare to other offline RL methods on a set of standard offline RL tasks, including learning from heterogeneous and homogeneous offline datasets? (2) How does LAPO compare to prior methods implemented with GMM policies when learning behaviors offline from heterogeneous datasets? and (3) In which setting does LAPO benefit from the latent policy training, versus only using the generative model? Can LAPO learn without constraining the latent values?

We study the first question by evaluating LAPO and comparing it to several prior methods on offline RL benchmarks with heterogeneous datasets. We consider a range of simulated robotic tasks, including navigation, locomotion, and manipulation, each with a corresponding static dataset with a heterogeneous data distribution. We also consider datasets with narrow and biased data distributions, containing near-optimal or random trajectories. To answer the second question, we include a comparison of AWAC [39] with a GMM policy, since AWAC trains using a similar advantage-weighted objective as the actor policy in LAPO. This comparison is only performed on heterogeneous datasets, since this is where we expect to see the most improvement from a multi-modal policy. Finally, the questions in (3), we conduct two ablation studies in which we train LAPO without learning the latent policy, i.e. evaluating only LAPO's action policy, and without limiting the latent values to the range $[-z_{\max}, z_{\max}]$. We compare this version of the method to the original LAPO on four offline RL tasks that differ in the amount of high-performing task-relevant data contained in the offline dataset. Our implementation is available at `https://github.com/pcchenxi/LAPO-offlienRL`.

## 6.1  Comparisons

We compare LAPO to 7 offline RL approaches: (1) vanilla behavioral cloning (BC), (2) the BCQ method [9], which approximates the dataset distribution with a generative model and manually selects the action with the maximum Q value among a set of generated actions, (3) the PLAS method [57], which learns a generative model via maximum likelihood and a latent-space policy to maximize the RL objective in the latent space, (4) the AWAC method [39], which performs advantage-weighted imitation learning with a forward-KL objective for the policy projection step, (5) the IQL method

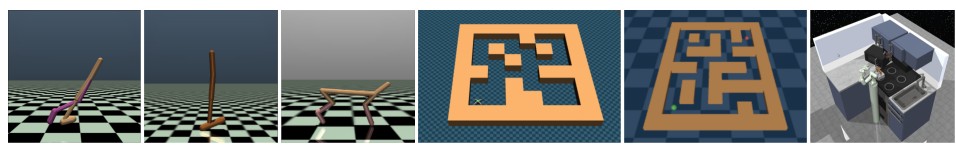

Figure 3: D4RL environments used in our evaluations: (from left to right) Walker 2D, Hopper, Halfcheetah, Antmaze, Maze2d, and Kitchen environments.

[23], which learns the value function using expectile regression without an explicit policy, and then extracts the policy using advantage weighted regression (AWR), (6) the CQL method [25] which learns a conservative Q-function, and (7) AWAC $_{(GMM)}$ which trains GMM policies using AWAC.

In our experiments, we train each method three times using three different random seeds each for one million steps, and report the return averaged over 10 test episodes from the trained policies. We used the implementation provided by [48] for the BCQ and AWAC, the original implementation for the PLAS, IQL and CQL, and our implementation for the BC method. We report the implementation details in the Appendix E.

## 6.2 Tasks and Datasets

We compare LAPO to the introduced prior methods on learning from both heterogeneous and homogeneous offline datasets. We evaluate the methods on three simulated task domains, locomotion, navigation, and manipulation domains (Figure 3), with heterogeneous data distributions, and also a locomotion task with random, narrow and biased data distribution. We consider two sources of heterogeneous datasets: (1) data collected by policies accomplishing different tasks, and (2) data collected by different policies that accomplish one task but in different ways. For (1), we use the multi-task heterogeneous datasets introduced by [54], and for (2), we leverage heterogeneous datasets from the standard D4RL benchmark [6]. More details about our tasks and datasets are provided in the Appendix F.

## 6.3 Experimental Results

**Heterogeneous datasets:** Table 2 reports the results of our experiments on heterogeneous datasets, and in Figure 6 (Appendix C), we present the learning performance of LAPO with 95% confidence interval on Heterogeneous tasks. Among the 12 tasks introduced in the previous section, LAPO (our method) achieves the best performance on nine tasks and the second-best performance on one task. Besides, on average, LAPO improves by 49% over the next best method on the heterogeneous datasets.

The heterogeneous locomotion tasks can be accomplished more easily by the prior methods since still 33% of the data comes from the target task. However, from the prior works, AWAC and CQL are the only methods that can accomplish the tasks. LAPO performs well on all tasks and on average yields the best performance on one out of the three tasks. This shows that LAPO is capable of learning from heterogeneity introduced by multi-task datasets.

The navigation tasks are in general more challenging especially for those with medium and large map sizes. The main challenge is to learn policies for long-horizon planning from datasets that do not contain optimal trajectories, and there are only very few states with rewards. LAPO significantly outperforms all prior works for all of the navigation tasks. Similarly, the manipulation tasks are also challenging, as they require the assembly of sub-trajectories related to completing a given task consisting of multiple sub-tasks. In addition, the agent has access to fewer training samples while having to learn to interact with complicated dynamics. LAPO outperforms previous approaches by a large margin on Kitchen-partial and Kitchen-mixed for which only a few optimal trajectories are provided by the dataset. It also performs competitively on the Kitchen-complete task and yields the second-best performance.

we observe that GMM policy training does not perform well on heterogeneous offline RL settings. The AWAC method with GMM policies yields similar or even worse performance compared to the original AWAC. These results suggest that the VAE is more robust to model the high-reward regions in heterogeneous data.

Table 2: The normalized performance of all methods on tasks with heterogeneous dataset. 0 represents the performance of a random policy and 100 represents the performance of an expert policy. The scores are averaged over the final 10 evaluations and 3 seeds. LAPO achieves the best performance on 9 tasks and achieves competitive performance on the rest 3 tasks. Results with 95%-confidence interval are reported in Appendix C.2

| Task ID | BC | BCQ | PLAS | AWAC | AWAC$_{(GMM)}$ | IQL | CQL | LAPO$_{(Ours)}$ |
|---|---|---|---|---|---|---|---|---|
| Walker2d-mix-forward-v1 | -5.65 | 0.91 | 22.91 | 71.89 | 78.09 | 28.25 | **102.75** | 74.17 |
| Walker2d-mix-backward-v1 | -84.56 | -1.37 | -27.58 | -7.95 | 71.33 | -46.25 | 66.64 | **99.22** |
| Walker2d-mix-jump-v1 | -72.92 | -41.43 | 0.56 | **51.08** | 28.01 | -46.41 | 37.28 | 43.20 |
| Maze2d-umaze-v1 | 0.99 | 18.91 | 80.12 | 94.53 | 19.45 | 51.00 | 22.86 | **118.86** |
| Maze2d-medium-v1 | 3.34 | 12.79 | 5.19 | 31.40 | 46.53 | 33.26 | 12.25 | **142.75** |
| Maze2d-large-v1 | -1.14 | 27.17 | 45.80 | 43.85 | 9.04 | 64.30 | 7.00 | **200.56** |
| Antmaze-umaze-diverse-v1 | 60.00 | 62.00 | 7.00 | 72.00 | 0.00 | 69.33 | 16.71 | **91.33** |
| Antmaze-medium-diverse-v1 | 0.00 | 11.33 | 8.67 | 0.33 | 0.00 | 73.00 | 1.00 | **85.67** |
| Antmaze-large-diverse-v1 | 0.00 | 0.67 | 1.33 | 0.00 | 0.00 | 48.00 | 11.89 | **61.67** |
| Kitchen-complete-v0 | 4.50 | 9.08 | 38.08 | 3.83 | 1.08 | **66.67** | 4.67 | 53.17 |
| Kitchen-partial-v0 | 31.67 | 17.58 | 27.00 | 0.25 | 0.42 | 32.33 | 0.55 | **53.67** |
| Kitchen-mixed-v0 | 30.00 | 11.50 | 29.92 | 0.00 | 3.92 | 49.92 | 1.86 | **62.42** |

Table 3: The normalized performance of all methods on locomotion tasks with random, narrow and biased dataset. 0 represents the performance of a random policy and 100 represents the performance of an expert policy. The scores are averaged over the final 10 evaluations and 3 seeds. LAPO achieves the best performance on 4 tasks and achieves competitive performance on the rest of the tasks. Results with 95%-confidence interval are reported in Appendix C.2

| Task ID | BC | BCQ | PLAS | AWAC | IQL | CQL | LAPO$_{(Ours)}$ |
|---|---|---|---|---|---|---|---|
| Hopper-random-v2 | 2.23 | 7.80 | 6.68 | 8.01 | 7.89 | 8.33 | **23.46** |
| Walker2d-random-v2 | 1.11 | 4.87 | **9.17** | 0.42 | 5.41 | -0.23 | 1.28 |
| Halfcheetah-random-v2 | 2.25 | 2.25 | 26.45 | 15.18 | 13.11 | 22.20 | **30.55** |
| Hopper-medium-v2 | 49.23 | 56.44 | 50.96 | 69.55 | 65.75 | **71.59** | 51.63 |
| Walker2d-medium-v2 | 47.11 | 73.72 | 76.47 | **84.02** | 77.89 | 82.10 | 80.75 |
| Halfcheetah-medium-v2 | 37.84 | 47.22 | 44.54 | 48.13 | 47.47 | **49.76** | 45.97 |
| Hopper-expert-v2 | 76.16 | 68.86 | 107.05 | **109.32** | **109.36** | 102.27 | 106.76 |
| Walker2d-expert-v2 | 79.22 | 110.51 | 109.56 | 110.46 | 109.93 | 108.76 | **112.27** |
| Halfcheetah-expert-v2 | 85.63 | 93.15 | 93.79 | 14.01 | 94.98 | 87.40 | **95.93** |

**Random/Narrow datasets:** Table 3 reports the results of the locomotion tasks with random, narrow and biased datasets. Our method yields the best performance for four out of nine tasks. It also achieves competitive performance on the rest of the tasks, showing that LAPO is applicable to general offline RL settings and is not limited to learning from heterogeneous datasets. However, the most significant gain is when it is applied to offline settings with heterogeneous datasets. Learning curves and results with 95%-confidence interval are reported in Appendix C.

## 6.4 Ablations

**Latent Policy Training**: To study the importance of the latent policy training in achieving good performances, we conduct an ablation study on 4 different tasks by eliminating the latent policy training from LAPO. The main motivation is that, as described in Section 4, the latent variable generative model is trained to incrementally learn the action distribution that lead to higher return, hence training the latent policy on top of that may seem redundant. In this case, another option for sampling latent variables is to sample from the prior distribution, which changes Equation 8 to:

$$V(s) = \mathbb{E}_{a \sim p_{\theta'}(\cdot|s,z) z \sim p(z)}[Q_{\phi'}(s, a)],$$

where, the latent $z$ is now sampled from a fixed prior.

As shown in Fig. 4, training a latent policy does not significantly contribute to higher performance for tasks such as the Walker2D-medium and Walker2d-mix-backward which contain sufficient task-

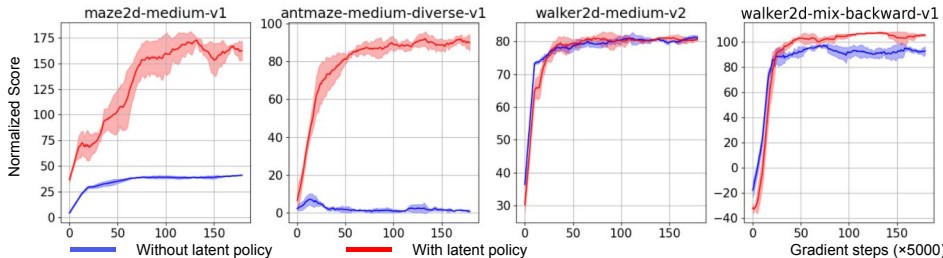

Figure 4: Comparison of LAPO with and without the latent policy on Maze2D-medium, Antmaze-medium-diverse, Walker2D-medium, and Walker2D-mix-backward task. The shaded area represents ±95%-confidence intervals, computed using 10 evaluations and 3 random seeds. For tasks with less optimal task-relevant data like Maze2D-medium, Antmaze-medium-diverse, the learned latent policy results in high-performance gains.

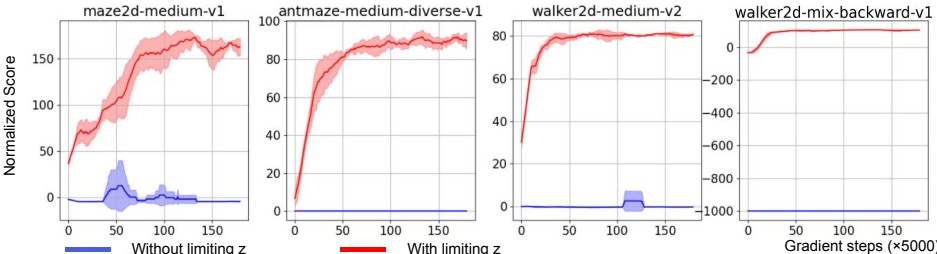

Figure 5: Comparison of LAPO with and without limiting the latent values. The policy performance is significantly worse when the latent values are not limited to the range $[-z_{\max}, z_{\max}]$.

relevant data. However, latent policy training results in large performance gains when learning from datasets which contain fewer optimal task-related data such as Maze2D and Antmaze. LAPO overcomes this challenge by explicitly optimizing the RL objective through learning the latent policy.

**Limiting the latent values**: We also study the importance of limiting the latent variable $z$ to be within the range $[-z_{\max}, z_{\max}]$. This is important to understand the limits of using the latent variable decoder as a generative model of in-distribution actions. Figure 5 illustrates the result of LAPO policy training using unbounded latent actions on four tasks. The policy training performance is significantly worse when we do not limit the latent values. This suggests that the the action policy can generate in-distribution actions only when in-distribution latent values (close to samples drawn from the prior distribution $p(z)$) are given as the input.

# 7 Conclusion

In this paper, we study an offline RL setup for learning from heterogeneous datasets where trajectories are collected using policies with different purposes, leading to a multi-modal data distribution, and in some cases, do not contain sufficient high-reward trajectories. Through empirical analysis, we find that in such cases, policies constrained methods may contain out-of-distribution actions and lead to suboptimal performance for continuous control tasks, especially for heterogeneous datasets. To address this challenge, we present the latent-variable advantage-weighted policy optimization (LAPO) algorithm, which learns a latent variable model that generates high-advantage actions when sampling from a prior distribution over the latent space. In addition, we train a latent policy that obtains state-conditioned latent actions which result in higher reward outcomes compared to sampling from the prior distribution. We compare our method to 6 prior methods on a variety of simulated locomotion, navigation, and manipulation tasks provided heterogeneous offline datasets, and also on standard offline RL benchmarks with narrow and biased datasets. We find that our proposed method consistently outperforms prior methods by a large margin on tasks with heterogeneous datasets, while being competitive on other offline RL tasks with narrow data distributions. For our future work, we will extend LAPO to multi-task offline RL settings in which an agent learns multiple RL tasks provided a heterogeneous dataset of diverse behaviors.

## Acknowledgments

This work is supported in part by Science and Technology Innovation 2030 – "New Generation Artificial Intelligence" Major Project (No. 2018AAA0100904) and National Natural Science Foundation of China (62176135).

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
