# LAPO: Latent-Variable Advantage-Weighted Policy Optimization for Offline Reinforcement Learning (Appendix)

## A  The temperature and the importance weight

We found empirically that the performance is sensitive to the choice of the temperature $\lambda$ when calculating the importance weight $\omega = \exp(A(s,a)/\lambda)$. A large $\lambda$ may constrain the policy to be too close to the behavior policy, leading to poor performance (Figure 2.b). While, if the $\lambda$ is too small, the computed importance weighted can be very large, which causes problems in the training process. Therefore, an additional tuning process is needed to find an appropriate $\lambda$ for different tasks.

Inspired by the work [23], in our implementation, we use a fixed importance weight $\omega^+ = 0.9$ for actions with positive advantage and use $\omega^- = 1 - \omega^+ = 0.1$ for actions with negative advantage. It simplifies the computation and avoids tuning $\lambda$ for different tasks. As we empirically observed, this technique yields good results for all settings.

**Effect of the advantage weight** To study the effect of the advantage weight $\omega$, we evaluate the learning performance on heterogeneous tasks using $\omega^+ \in \{0.5, 0.7, 0.9, 0.99\}$. In our implementation, we set the importance weight to actions with positive advantages as $\omega^+$ and set actions with negative advantages as $\omega^- = 1 - \omega^+$. When $\omega^+ = 0.5$, all actions are equally weighted, and the method becomes identical to PLAS, which shares the same network structure and data processing techniques as LAPO. As $\omega$ increases, more weights are added to the actions with positive advantage, and when $\omega^+ = 0.99$, the generative model tends to ignore the actions with negative advantage. The results are presented in Table 4. We observe a significant performance improvement when we increase $\omega^+$ over 0.5. This result suggests that the dynamic re-weighting mechanism plays an important role in the algorithm, making LAPO outperform PLAS when learning from heterogeneous data sets.

Table 4: The normalized performance and 95%-confidence interval on tasks with heterogeneous dataset using different advantage weights. 0 represents the performance of a random policy and 100 represents the performance of an expert policy. The scores are averaged over the final 10 evaluations and 3 seeds.

| Task ID | $\omega$=0.5 | $\omega$=0.7 | $\omega$=0.9 | $\omega$=0.99 |
|---|---|---|---|---|
| Walker2d-mix-forward-v1 | 42.86 ± 34.77 | 71.28 ± 16.8 | **74.17 ± 22.64** | 61.2 ± 26.94 |
| Walker2d-mix-backward-v1 | 85.65 ± 2.67 | 98.76 ± 1.41 | 99.22 ± 5.19 | **103.35 ± 1.31** |
| Walker2d-mix-jump-v1 | 2.87 ± 30.08 | 38.12 ± 4.48 | 43.2 ± 3.8 | **44.02 ± 10.79** |
| Maze2d-umaze-v1 | 58.3 ± 23.41 | **128.63 ± 24.49** | 118.86 ± 55.66 | 108.2 ± 37.85 |
| Maze2d-medium-v1 | 92.74 ± 22.85 | 123.37 ± 20.98 | 142.75 ± 11.67 | **148.11 ± 24.58** |
| Maze2d-large-v1 | 133.35 ± 15.32 | 181.95 ± 40.83 | **200.56 ± 18.86** | 146.76 ± 43.82 |
| Antmaze-umaze-diverse-v1 | 89.33 ± 9.12 | 86.67 ± 8.54 | 91.33 ± 10.67 | **94.67 ± 1.07** |
| Antmaze-medium-diverse-v1 | 66.67 ± 21.66 | **86.33 ± 7.0** | 85.67 ± 9.24 | 83.33 ± 9.3 |
| Antmaze-large-diverse-v1 | 50.33 ± 13.62 | 63.33 ± 6.49 | 61.67 ± 21.34 | **65.33 ± 19.76** |
| Kitchen-complete-v0 | 51.83 ± 5.94 | 52.67 ± 6.15 | **53.17 ± 9.62** | 52.67 ± 7.07 |
| Kitchen-partial-v0 | 46.67 ± 2.54 | 49.83 ± 3.71 | **53.67 ± 12.22** | 51.33 ± 16.33 |
| Kitchen-mixed-v0 | 52.13 ± 3.08 | 55.5 ± 4.55 | **62.42 ± 7.06** | 60.33 ± 6.7 |

## B  Connections to PLAS

LAPO is similar in structure to PLAS [57] as they both learn a generative model and have a policy over the latent space. However, the generative model in PLAS is pre-trained to approximate the distribution over the entire data and is fixed when training the latent policy. The expressiveness of the model on high-return area can be limited if the number of such samples is small in the dataset. In contract, the generative model of LAPO is trained to selectively represent actions that lead to higher returns based on their current advantage values. For example, in the toy experiment in Figure 2, the dataset contains 9% of actions that traverse through the high reward area. When drawing 1,000 z from the prior $p(z)$ of the latent space learned by PLAS, only **4%** of the samples are decoded as high-return actions, while in LAPO, **45%** of the decoded actions are high-return actions. We also demonstrate the importance of the dynamic re-weighting mechanism in Table 4.

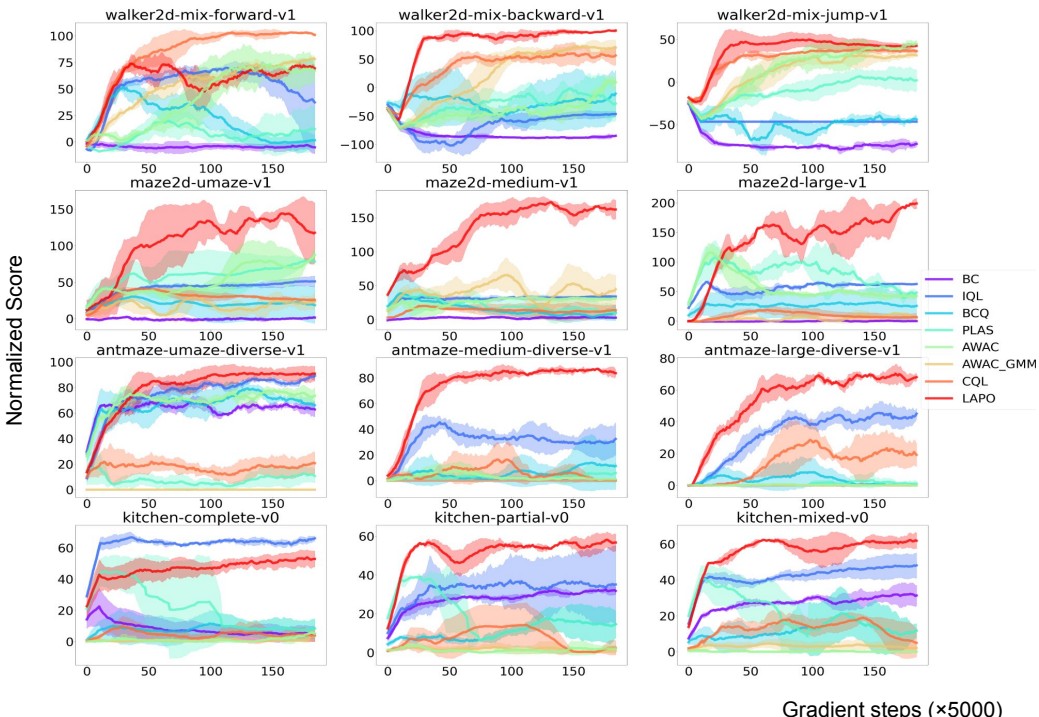

Figure 6: The learning curve of LAPO with 95% confidence interval for heterogeneous tasks. The x-axis is the training epochs, each containing 5,000 gradient updates, and the y-axis is the normalized score. For better visualization, the scores are smoothed by a window with length 20.

## C    Additional information on experimental results

### C.1    Learning curves

We plot the learning curves of heterogeneous tasks in Figure 6, and plot the learning curves of random, narrow and bias tasks in Figure 7.

### C.2    Results with confidence interval

Table 6 and 7 report our experimental results with 95%-confidence interval on heterogeneous datasets and random, narrow and bias datasets, respectively.

### C.3    Examples trajectories of the navigation tasks

To visualize the performance of the policy learned by LAPO, in figure 8, we plot ten trajectories using the learned policy in the six navigation tasks.

## D    Avoiding out-of-distribution action using the VAE decoder

In theory, the objective of VAE is to maximize the log-likelihood of data in the training dataset given a latent value $z$ sampled from a prior distribution $p(z)$. Therefore, it is unlikely to generate out-of-distribution samples from an optimal decoder trained in VAE given in-distribution $z$ inputs. To verify the practical performance of the VAE decoder on avoiding out-of-distribution samples in the heterogeneous setting, we conduct an experiment to train a VAE with a 2-dimensional latent space to approach a mixture distribution of two Gaussians: $0.5N(-10,1) + 0.5N(10,1)$. We draw 5,000 $z$ values from a truncated unit Gaussian distribution limited to [-2, 2], and decode these values using the trained decoder. We plot the histogram of the generated samples by the VAE, and the input latent value $z$ in Figure 9.

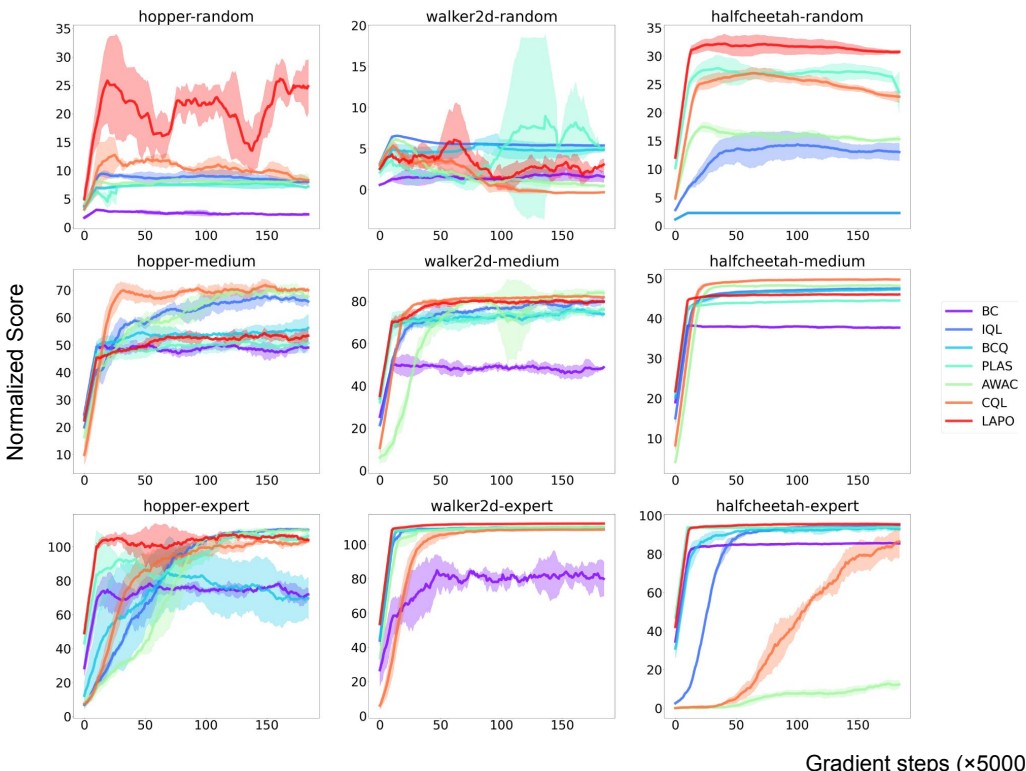

Figure 7: The learning curve of LAPO with 95% confidence interval for random, narrow and bias tasks. The x-axis is the training epochs, each containing 5,000 gradient updates, and the y-axis is the normalized score. For better visualization, the scores are smoothed by a window with length 20.

in Figure 9, we observe that there are only 19 samples out of 5,000 located between [-5, 5], which should be considered as out-of-distribution to the given training distribution. The $z$ values that generate these out-of-distribution actions lie in a line in the latent space between the two areas associated with the two modes. We confirm that, in practice, it is possible to generate out-of-distribution actions from the decoder, however, most of the generated samples are in-distributed. According to our experimental result on heterogeneous tasks in Table 2, we argue that the small number of potential out-of-distribution actions in the latent space have minor effects to our method.

## E   Implementation Details

### E.1   Implementation of prior methods

The BCQ and AWAC methods are based on the implementations of *d3rlpy*: `https://github.com/takuseno/d3rlpy`, and PLAS, IQL and CQL methods are implemented using the original implementations provided by the authors of the papers: `https://github.com/Wenxuan-Zhou/PLAS`, `https://github.com/ikostrikov/implicit_q_learning`, and `https://github.com/aviralkumar2907/CQL`. For AWAC-GMM method, we extend the policy class implemented in *d3rlpy* AWAC from a single Gaussian policy to 5-head GMM policy. The implementation of LAPO is available at `https://github.com/pcchenxi/LAPO-offlienRL`.

We used the same hyperparameters as the original paper or the code provided by the author for the prior methods. All models are trained using a NVIDIA P100 GPU.

### E.2   Network Hyperparameters

The hyper-parameters used in our experiment are summarized in Table 5. We use $\beta = 0.3$ on Maze2D tasks and $\beta = 1$ for other tasks. We set the dimension of the latent policy to be two times larger than

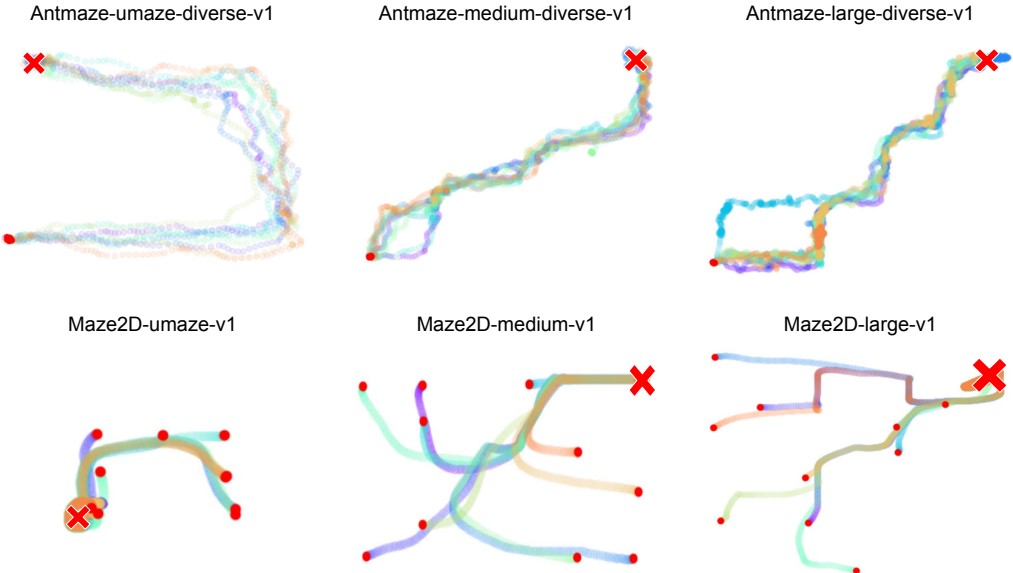

Figure 8: Trajectories learned by LAPO in the navigation tasks. The red dots indicate the starting states of each trajectories, and the red cross indicates the target state. In antmaze environments, the agents are initialized at the same xy position but with different joint values. In maze2D environments, the agents are initialized at different locations.

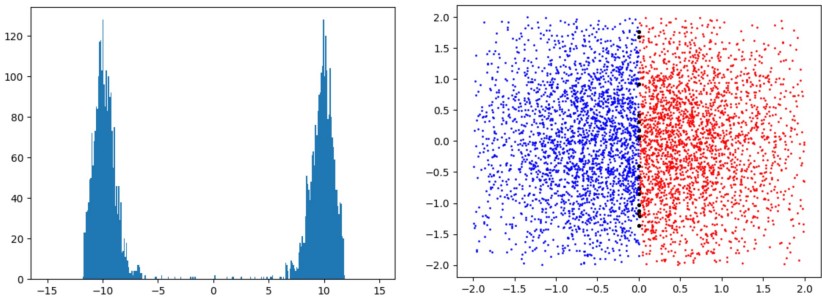

Figure 9: (left) The histogram of generated samples of a VAE model, which is trained to approximate a mixture distribution of two Gaussians: $0.5N(-10, 1) + 0.5N(10, 1)$. (right) Plot of the input 2-dimensional latent values that are used to generate samples. The blue points are decoded as samples for the left mode, the red points are decoded as samples for the right mode, and the black points in the middle are decoded as samples between [-5, 5]. There are only 19 samples out of 5,000 located between [-5, 5] that can be considered as out-of-distribution to the training data.

the dimension of the action space. We truncate the output of the latent policy to stay in the range of $[-2.0, 2.0]$. In TD3, we use the clipped double Q-learning ([8]) to compute the state value by combining the outputs of two Q-functions. For Antmaze tasks, we combine them as 0.7*min(Q1, Q2) + 0.3*max(Q1, Q2); for the rest of the tasks, we take the minimum of the two Q values.

### E.3 Data pre-processing

We normalized the observations and actions in the dataset for all tasks. For Antmaze, we multiplied Antmaze's reward by 100. For the rest of the tasks, we divided the reward by the maximum reward in the dataset.

Table 5: Hyper-parameters and network architecture for training LAPO

|  | Hyperparameter | Value |
|---|---|---|
| TD3 & LAPO Hyperparameters | Optimizer | Adam |
|  | Critic learning rate | 0.0002 |
|  | Action policy learning rate | 0.0002 |
|  | Latent policy learning rate | 0.0002 |
|  | Mini-batch size | 512 |
|  | Discount factor | 0.99 |
|  | Target update rate | 0.005 |
|  | Policy noise | 0.1 |
|  | Policy noise clipping | 0.3 |
|  | VAE $\beta$ | 0.3 for Maze2D, 1.0 for other tasks |
|  | Importance weight $\omega$ | 0.9 for positive adv, 0.1 for negative adv (A) |
|  | Latent action size | $2 \times |A|$ |
|  | Latent action limit | [-2.0, 2.0] |
| Architecture | Critic hidden layers | [256, 256, 256] |
|  | Action policy hidden layers | [256, 256, 256] |
|  | Latent policy layers | [256, 256, 256] |
|  | Vae encoder layers | [256, 256, 256] |
|  | Activation function | ReLU |

## F   Tasks and Datasets

**Locomotion:** We adopt the task settings introduced in [54] to define three locomotion tasks. The datasets are constructed using all of the training data in the replay buffer of three separate policy training sessions each trained for $0.5$ million steps using the SAC method [14]. The tasks are to control a Walker2D agent to run forward, backward, and jump. Similar to the original work, we keep a single replay buffer with all of the transitions of all of the three tasks, and form three offline datasets by relabeling the rewards using the reward function provided for each task. We refer to the datasets as *Walker2d-mix-forward*, *Walker2d-mix-backward* and *Walker2d-mix-jump* in the rest of this section.

**Navigation:** We use the two datasets *Maze2d-sparse* and *Antmaze-diverse* from the D4RL benchmark. The trajectories in the datasets are collected by training goal-reaching policies to navigate to random goals from random initial positions. Provided the pre-collected trajectories, the rewards are relabeled to generate offline data to navigate to different goal positions. Therefore, the task is to learn from data generated by policies that try to accomplish different tasks not aligned with the task at hand. The target task has a sparse binary reward function which gives a reward of *one* only when the agent is close to the goal position, and *zero* otherwise.

**Manipulation:** For the manipulation domain, we leverage the FrankaKitchen task from the D4RL benchmark. The task is to control a 9-DoF Franka robot to manipulate common household items such as microwave, kettle, and oven, in sequence to reach desired target configuration for several items. There are three datasets collected by human demonstrations: (1) *Kitchen-complete* which consists of successful trajectories that perform tasks in order, (2) *Kitchen-partial* which similar to (1) consists of some successful task-relevant trajectories, but also contains unrelated trajectories that are not necessarily related to reach any target configurations, and (3) *Kitchen-mixed* which consists of partial trajectories that do not solve the entire task, and requires the highest level of generalization from the agent to accomplish the task. The kitchen environment has a sparse reward that is provided whenever an item is at its target configuration.

**Locomotion (narrow/random data distribution):** For the offline RL task of learning from narrow data distributions, we leverage three Gym-MuJoCo locomotion tasks from the D4RL benchmark with random, narrow and biased data distribution: *Hopper*, *Walker2d* and *Halfcheetah*. Each task contains three datasets collected using a randomly initialized policy ("-random"), a semi-trained policy ("-medium"), and a fully trained policy ("-expert"), respectively. The behavioral policies are trained online using the SAC method.

| Task ID | BC | BCQ | PLAS | AWAC | AWAC (GMM) | IQL | CQL | LAPO(Ours) |
|---|---|---|---|---|---|---|---|---|
| Walker2d-mix-forward-v1 | -5.65 ± 6.06 | 0.91 ± 38.43 | 22.91 ± 7.45 | 71.89 ± 11.63 | 78.09 ± 0.93 | 28.25 ± 6.87 | 102.75 ± 3.44 | 74.17 ± 22.64 |
| Walker2d-mix-backward-v1 | -84.56 ± 0.57 | -1.37 ± 9.48 | -27.58 ± 28.29 | -7.95 ± 20.63 | 71.33 ± 28.98 | -46.25 ± 14.93 | 66.64 ± 32.08 | 99.22 ± 5.19 |
| Walker2d-mix-jump-v1 | -72.92 ± 23.35 | -41.43 ± 0.02 | 0.56 ± 16.92 | 51.08 ± 17.38 | 28.01 ± 8.23 | -46.41 ± 15.03 | 37.28 ± 3.36 | 43.2 ± 3.8 |
| Maze2d-umaze-v1 | 0.99 ± 7.01 | 18.91 ± 14.89 | 80.12 ± 29.12 | 94.53 ± 5.93 | 19.45 ± 16.91 | 51.0 ± 10.88 | 22.86 ± 3.1 | 118.86 ± 55.66 |
| Maze2d-medium-v1 | 3.34 ± 8.37 | 12.79 ± 1.9 | 5.19 ± 15.1 | 31.4 ± 16.2 | 46.53 ± 8.01 | 33.26 ± 23.9 | 12.25 ± 12.25 | 142.75 ± 11.67 |
| Maze2d-large-v1 | -1.14 ± 5.42 | 27.17 ± 5.45 | 45.8 ± 26.26 | 43.85 ± 15.87 | 9.04 ± 14.92 | 64.3 ± 6.68 | 7.0 ± 6.26 | 200.56 ± 18.86 |
| Antmaze-umaze-diverse-v1 | 60.0 ± 16.0 | 62.0 ± 5.33 | 7.0 ± 5.33 | 72.0 ± 37.34 | 0.0 ± 5.33 | 85.67 ± 0.0 | 16.71 ± 11.14 | 91.33 ± 10.67 |
| Antmaze-medium-diverse-v1 | 0.0 ± 0.0 | 11.33 ± 21.34 | 8.67 ± 26.67 | 0.33 ± 5.33 | 0.0 ± 10.67 | 9.0 ± 0.0 | 1.0 ± 5.33 | 85.67 ± 9.24 |
| Antmaze-large-diverse-v1 | 0.0 ± 0.0 | 0.67 ± 9.24 | 1.33 ± 0.0 | 0.0 ± 0.0 | 0.0 ± 0.0 | 33.67 ± 0.0 | 11.89 ± 24.45 | 61.67 ± 21.34 |
| Kitchen-complete-v0 | 4.5 ± 1.33 | 9.08 ± 5.81 | 38.08 ± 6.67 | 3.83 ± 6.67 | 1.08 ± 4.0 | 66.67 ± 5.33 | 4.67 ± 2.67 | 53.17 ± 9.62 |
| Kitchen-partial-v0 | 31.67 ± 4.0 | 17.58 ± 19.37 | 27.0 ± 2.31 | 0.25 ± 2.31 | 0.42 ± 0.0 | 32.33 ± 0.0 | 0.55 ± 6.1 | 53.67 ± 12.22 |
| Kitchen-mixed-v0 | 30.0 ± 9.62 | 11.5 ± 10.59 | 29.92 ± 6.93 | 0.0 ± 6.93 | 3.92 ± 0.0 | 49.92 ± 1.33 | 1.86 ± 10.67 | 62.42 ± 7.06 |

Table 6: The normalized performance of all methods on tasks with heterogeneous dataset. 0 represents the performance of a random policy and 100 represents the performance of an expert policy. The scores are averaged over the final 10 evaluations and 3 seeds, ± the 95%-confidence interval. LAPO achieves the best performance on 9 tasks and achieves competitive performance on the rest 3 tasks.

| Task ID | BC | BCQ | PLAS | AWAC | IQL | CQL | LAPO$_{(Ours)}$ |
|---|---|---|---|---|---|---|---|
| Hopper-random-v2 | 2.23 ± 0.18 | 7.8 ± 0.04 | 6.68 ± 0.04 | 8.01 ± 0.04 | 7.89 ± 1.69 | 8.33 ± 1.29 | 23.46 ± 0.62 |
| Walker2d-random-v2 | 1.11 ± 1.36 | 4.87 ± 0.13 | 9.17 ± 0.28 | 0.42 ± 0.26 | 5.41 ± 0.39 | -0.23 ± 0.12 | 1.28 ± 2.14 |
| Halfcheetah-random-v2 | 2.25 ± 0.0 | 2.25 ± 2.56 | 26.45 ± 0.0 | 15.18 ± 0.67 | 13.11 ± 0.82 | 22.2 ± 1.36 | 30.55 ± 0.21 |
| Hopper-medium-v2 | 49.23 ± 3.74 | 56.44 ± 5.16 | 50.96 ± 4.27 | 69.55 ± 0.45 | 65.75 ± 4.93 | 71.59 ± 10.3 | 51.63 ± 3.27 |
| Walker2d-medium-v2 | 47.11 ± 2.09 | 73.72 ± 6.38 | 76.47 ± 10.61 | 84.02 ± 1.76 | 77.89 ± 2.45 | 82.1 ± 6.26 | 80.75 ± 0.83 |
| Halfcheetah-medium-v2 | 37.84 ± 0.17 | 47.22 ± 0.12 | 44.54 ± 0.43 | 48.13 ± 0.39 | 47.47 ± 0.76 | 49.76 ± 0.54 | 45.97 ± 0.32 |
| Hopper-expert-v2 | 76.16 ± 11.27 | 68.86 ± 2.19 | 107.05 ± 11.64 | 109.32 ± 3.41 | 109.36 ± 2.33 | 102.27 ± 7.5 | 106.76 ± 3.6 |
| Walker2d-expert-v2 | 79.22 ± 7.22 | 110.51 ± 0.07 | 109.56 ± 0.62 | 110.46 ± 0.34 | 109.93 ± 0.31 | 108.76 ± 0.03 | 112.27 ± 0.08 |
| Halfcheetah-expert-v2 | 85.63 ± 0.3 | 93.15 ± 0.17 | 93.79 ± 7.39 | 14.01 ± 0.68 | 94.98 ± 0.82 | 87.4 ± 20.2 | 95.93 ± 0.22 |

Table 7: The normalized performance of all methods on locomotion tasks with random, narrow and biased dataset. 0 represents the performance of a random policy and 100 represents the performance of an expert policy. The scores are averaged over the final 10 evaluations and 3 seeds, ± the 95%-confidence interval.