# OpenReview forum: "LAPO: Latent-Variable Advantage-Weighted Policy Optimization for Offline Reinforcement Learning"
_NeurIPS.cc/2022/Conference — NeurIPS 2022 Accept_

### Official Review · Reviewer_cCVP · 2022-06-20

**Rating:** 4
**Confidence:** 4
**Soundness:** 2 fair
**Presentation:** 2 fair
**Contribution:** 2 fair

**Summary:**

In offline RL, the static datasets are often heterogeneous.
The standard policies constrained methods can be insufficient to learn from such datasets, since they may artificially assign high probability to out-of-distribution actions, leading to sub-optimal policies.
To address this challenge, this paper proposes to learn a latent variable model that generates high-advantage actions when sampling from a prior distribution on the latent space.
Furthermore, this paper trains a latent policy to obtain state-conditional latent actions, which empirically results in better outcomes compared to sampling from a fixed prior distribution.
The proposed method, LAPO, performs well on several tested datasets.

**Questions:**


1. L30-31: why learning from heterogeneous settings have the problem of "lack of sufficient high-return trajectories?" Why the dataset "is constructed by re-labeling state-action pairs from some different tasks?" I don't think the tested Maze2d and Antmaze datasets has this relabelling process.
2. It seems from Figure 2 that AWAC_GMM perform competitive to LAPO_VAE? Could you provide some intuition why it underperforms the LAPO in many of the D4RL datasets?
3. L60: How to "construct a state-conditioned latent space that represents high-return actions" if at some state the dataset only contain low-return actions? This can be exacerbated when the dataset only contains few high-return actions, which is what this paper wants to address.
4. For a given targeted task, if the goal is to "select which (of the multiple conflicting action) samples to use," why bother using a "more expressive policy class to represent the multi-modal action distributions?" Since we only need one action choice at each state, why couldn't we just use a deterministic policy + advantage weighted learning? \
As a further discussion, this paper aims at the following two goals: **(1)** represent the multi-modal action distributions, where the multiple modes come from the distribution of conflicting actions; and **(2)** learn a mechanism to select which samples (actions) to use. These two goals themselves seems conflicting and I am confused why we would like to simultaneously achieve both goals. In particular, since this paper use advantage weighted learning, the authors seem to implicitly assume that one action-mode is significantly better the others. Then why do we want to model other inferior action modes?
5. L113-114: why the latent values will be decoded into in-distribution actions? Why couldn't the decoder generate out-of-distribution actions?
6. I am confused by this sentence in L114-116 "The parameter $\theta'$ denotes the parameter of the decoder in earlier versions, which do not provide gradient information and therefore do not interfere with the optimization of the first component."\
If the $p_{\theta'}$ network does not provide gradient information, how can we train the $\pi_v$ network? \
A follow up discussion, the use of $a$ in the first term of Eq. (3) seems unnecessary due the use of $\theta'$. Can we define the Q function as $Q(s,z)$ so that the gradient information can be back-propagated directly to the $\pi_v$ network? I think this one has better semantic meaning that the latent policy, which maximizes the Q-function, does not rely on the action decoder during training. The first term in Eq. (3) is confusing to me since the training of $\pi_v$, and hence the generation of the latent values, entwists with the (lagged) decoder. Will the latent policy change as the decoder change? If so, could you explain the intuition behind this design? \
A even further discussion, since the decoder training already favors high advantage actions, I am not sure if the latent policy can actually *learn* "latent actions which result in high returns after being converted to the original action space using the decoder", as claimed in L148-149.
7. L119-120, the sentence "while actively prioritize regions ... through the importance weights" is confusing. As shown in Eq. (3), the training of the latent policy $\pi_v$ does not involve the "importance weight". Could you explain this?
8. L120-121, the sentence "Using the decoder, the latent policy will naturally avoid sampling OOD actions" is confusing. Why the the OOD actions can be *naturally* avoided? How do you define the OOD actions in the latent space, the (function) range of the latent policy?
9. I think Eq. (4) is wrong. Could you double check it with the Eq. (4) of PLAS [1]?
10. I am still unsure how the estimation of the advantage function can be exempted from the OOD actions? Since the Q-function is trained by the TD-error Eq. (2), a wield $a'$ can propagate the error. If so, the training of the decoder $p_\theta$ will be affected by the inaccurate advantage function.

[1] Zhou, Wenxuan, Sujay Bajracharya, and David Held. "PLAS: Latent action space for offline reinforcement learning." arXiv preprint arXiv:2011.07213 (2020).

**Limitations:**

The authors briefly discuss the limitations but not potential negative societal impacts.


**Strengths And Weaknesses:**

### Strengths
1. It is interesting to see a method to learn a  policy more flexible than the standard deterministic / Gaussian policy.

2. The empirical results are promising.

### Weaknesses
1. Novelty: while the whole algorithm is novel, it seems to be a combination of ideas in PLAS and AWAC. Thus, the novelty is incremental compared with prior work.

2. Presentation: explanation to the proposed method is insufficient. In particular, it will be beneficially to explain the intuition/motivation behind the design of each component. \
Please check the ***Questions*** section for questions I have for this paper.

---

> ### Author Response · Authors · 2022-08-02
> **Response to Reviewer cCVP (1)**
>
>
> Thanks for your constructive and detailed feedback. We address the comments below.
>
> **Q1:**
> There is no guarantee of having sufficient high-return trajectories after re-labeling a pre-collected dataset with a novel reward function unknown to the policy used to collect the data. This is the case for our heterogeneous locomotion dataset in which offline data from different locomotion tasks (run forward, backward, and jump) are separately re-labeled to be used for each task. Here, we expect the transitions for running forward helps when learning to jump, even though running trajectories does not result in high returns for the jumping task.
>
> The heterogeneous dataset contains a mixture of trajectories generated by policies trained for various tasks. As a result, the number of samples relevant to a particular task is small.
> For example, the D4RL Maze2D and Antmaze "-diverse" datasets used in the experiments were collected by commanding the agent to go to multiple random target locations. Each state-action pair in the dataset was then re-labeled by a binary reward for a given fixed target, and only states that are close enough to the given target would have positive rewards.
>
>
> **Q2:**
> It is practically challenging to train GMM policies, especially for more demanding RL tasks. GMM initialization considerably affects the training performance. For the toy task in Figure 2, we manually initialized the GMM with the two modes of the action distribution. However, for more complicated tasks, e.g., D4RL tasks, such priors do not exist.
>
> Furthermore, as our ablation study shows, training the latent policy of the LAPO significantly improves the performance compared to just using the VAE model to learn high-return trajectories. Therefore, LAPO's performance is not only attributed to the use of a more expressive policy architecture, and this is another reason why LAPO outperforms AWAC\_GMM in the more complicated settings.
>
>
> **Q3:**
> LAPO's goal is to stitch trajectories to find a path to high-return states. Let's assume that our offline dataset includes mostly low-return state-action trajectories but also a few high-return transitions. LAPO learns to stitch state-action transitions by learning a generative model (decoder) that, in the beginning, tries to find all possible actions for every state in the dataset. At each iteration, the generative model assigns higher distributions over actions with positive advantages even though those state-action samples are not very frequent in the dataset. By alternating between (1) updating the advantages of state-action pairs in the dataset, and (2) updating the generative model to generate actions with positive advantages, LAPO can stitch state-action transitions to find a path to high-return transitions.
> An important difference from other conservative offline RL algorithms is that conservatism in LAPO does not prevent it from learning high-return but low-density state-action samples.
>
> **Q4:**
> We confirm that the ultimate goal is to learn a deterministic policy that performs optimally for a given task. However, converging to such a policy requires an expressive stochastic policy that allows us to update the multi-modal action distribution which we begin with (assuming a heterogeneous setting), and revise the distribution in every training step until we converge to a deterministic policy.
> LAPO does this by constructing a latent space by alternating between updating the advantages and learning to generate high-advantage actions.
> This design choice is critical for the learning algorithm to converge.
>
>
> **Q5:**
> The VAE is trained to maximize the log-likelihood of actions in the given dataset. Given a fully trained VAE model for which the VAE loss is minimized, the decoder should produce in-distribution samples when in-distribution states and latent values are given as inputs. The states come from the offline dataset, and the latent samples are limited to stay in distribution when querying the generative model. Please see our ablation study ("limiting the latent values" L296), in which we show the effect of sampling out-of-distribution latent variables on the policy training performance.
> Prior works on VAE training also confirm our view that VAEs models are unlikely to generate OOD samples for in-distribution inputs. It is suggested in [3] (Can VAEs Generate Novel Examples? Neurips workshop 2018) and [4] (Amortized Inference Regularization NeurIPS 2018) that the reconstruction obtained from an optimal decoder of a VAE is a convex combination of examples in the training data. Therefore, the decoder is unlikely to generate OOD samples for in-distribution inputs.

---

> > ### Author Response · Authors · 2022-08-02
> > **Response to Reviewer cCVP (2)**
> >
> >
> > **Q6:**
> > We have revised L108-123 (L129-142 in the revised version) of the method section to improve clarity.
> > $\theta'$ is the parameter of the decoder in the previous iteration, which is fixed when updating the latent policy. The purpose of using $\theta'$ is to make sure that the training of the decoder $p_\theta$ is not affected by the latent policy updating.
> >
> > We cannot stabilize the training of $Q(s, z)$ since the generative model is updated in every iteration; and as the result, the same latent value $z$ will be mapped to different actions after each iteration.
> >
> > The latent policy, conditioned on an observed state, outputs a latent value (latent action) that, using the generative model (decoder), will be decoded to an action that maximizes the Q values.
> >
> >
> > **Q7:**
> > We thank the reviewer for bringing this up.
> > We revised the sentences, and removed the texts that are confusing.
> >
> >
> > **Q8:**
> > Given a state from the training dataset, and sampling a latent value $z$ from the prior $p(z)$, the generative model (decoder) $p(a|s, z)$ is limited to only generating in-distribution actions. As discussed earlier, the VAE decoder cannot generate novel actions not seen in the training dataset given that the inputs are in-distribution.
> >
> >
> > **Q9:**
> > We use a beta-VAE to learn the weighted distribution. Compared to equation (4) in [1], we have an additional $\beta$ to balance the reconstruction and the KL-loss, and an addition $\omega$, which is the importance weight computed using the action advantages. The prior distribution $p(z)$ is a unit Gaussian that is independent of the state.
> >
> >
> > **Q10:**
> > According to [4] (Off-Policy Deep Reinforcement Learning without Exploration, ICML 2019), The Q function can be fairly evaluated if the policy only traverses transitions contained in the dataset.
> > Therefore, as long as the decoder generates in-distribution actions, the extrapolation error of the Q values is mitigated, and the advantage estimation can be done as accurate as possible.

---

> > > ### Comment · Reviewer_cCVP · 2022-08-06
> > > **Response to the authors**
> > >
> > >
> > > Dear authors,
> > >
> > > Thank you for your detailed responses. While several of my previous questions/concerns have been addressed, there are still outstanding ones.
> > >
> > > > Therefore, LAPO's performance is not only attributed to the use of a more expressive policy architecture, and this is another reason why LAPO outperforms AWAC_GMM in the more complicated settings.
> > >
> > > I am a little confused about this sentence. I think both LAPO and AWAC_GMM use advantage-weighting scheme? Can the author explain further the other advantage of LAPO over AWAC_GMM except using a more advanced policy architecture?
> > >
> > > > LAPO's goal is to stitch trajectories to find a path to high-return states.
> > >
> > > Can the author point to the location in the paper where this goal is stated and explained?
> > >
> > > > LAPO learns to stitch state-action transitions by learning a generative model (decoder) that, in the beginning, tries to find all possible actions for every state in the dataset.
> > >
> > > I don't think "find all possible actions for every state in the dataset" is possible, especially when the offline dataset has limited size or has limited state-action distribution.
> > >
> > > > By alternating between (1) ..., and (2) ..., LAPO can stitch state-action transitions to find a path to high-return transitions.
> > >
> > > I don't quite understand why *(1)..., and (2)...* can "**stitch** state-action transitions to find a path to high-return transitions".
> > >
> > > > We confirm that the ultimate goal is to learn a deterministic policy that performs optimally for a given task ... until we converge to a deterministic policy
> > >
> > > Do the authors have empirical evidence showing that the learned policy does converge to a deterministic policy in order to support the claim here?
> > >
> > > Additionally, since the ultimate goal is to learne a deterministic policy, why couldn't we just use "deterministic policy + advantage weighted training"?
> > >
> > > > It is suggested in [3] .. and [4] ... that the reconstruction obtained from an optimal decoder of a VAE is a convex combination of examples in the training data.
> > >
> > > This actually coincides with my concern on the mode-covering behavior of VAE, which is also pointed out by Reviewer eXjC.
> > >
> > > Consider a simple example where the target distribution is a mixture of two Gaussians $0.5 \cdot N(-10, 1) + 0.5 \cdot N(10, 1)$. The training data obviously centers around the modes $-10$ and $10$.
> > > But the obtained reconstruction as "a convex combination of examples in the training data" can be, e.g., $0$, on which the target distribution has low density.
> > > I think such low density samples are considered as OOD.
> > >
> > > This is especially a concern since this paper aims at heterogeneous/multi-modal datasets.
> > >
> > > > We cannot stabilize the training of $Q(s,z)$, ..., the same latent value $z$ will be mapped to different actions after each iteration.
> > >
> > > This is actually my concern on the interplay between the latent policy and the decoder.
> > > The training mechanism of the proposed method remains unclear to me.
> > >
> > > Please refer to my original review for details on my such concerns.

---

> > > > ### Author Response · Authors · 2022-08-09
> > > > **Response to Reviewer cCVP (1)**
> > > >
> > > >
> > > > Thank you for your detailed feedback. We address the comments below.
> > > >
> > > > **Q1: Can the author explain further the other advantage of LAPO over AWAC\_GMM except using a more advanced policy architecture?**
> > > >
> > > >
> > > > Both LAPO and AWAC\_GMM use advantage-weights to learn a policy (a generative model in LAPO's case). They both use expressive policies to represent multi-modal action distributions to avoid OOD actions. However, an important advantage of LAPO over AWAC\_GMM is the latent policy, which is trained on top of the generative model, and explicitly selects optimal actions (w.r.t. the Q-function) from the high-advantage actions that are generated by the generative model (Equation 9 in the revised paper). Please see our ablation studies (Figure 4) for a comparison between LAPO with and without the latent policy training. In contrast, the AWAC\_GMM does not have this mechanism to select an optimal action from high-advantage actions learned by the policy. As a result, the best performance we may expect from AWAC\_GMM is LAPO without the latent policy, which is significantly worse than the original LAPO (Figure 4). This has also been shown by our toy navigation task in Table 1 and Figure 2 demonstrating that training the latent policy significantly improves the performance.
> > > >
> > > >
> > > > **Q2: Questions related to "stitch trajectories to find a path to high-return states". And
> > > > Q3 in the original comments: How to "construct a state-conditioned latent space that represents high-return actions" if at some state the dataset only contain low-return actions?**
> > > >
> > > > We used the term "stitching trajectory" in reply to Q3 in the original comments to refer to the learning process of selecting actions that lead to high-return states from the current state.
> > > > We clarify our answer to Q3 as below.
> > > >
> > > > The expected return of a state-action pair is estimated by the Q-function, which is updated iteratively. At the beginning of the training, all state-action pairs have the same advantage, and therefore, all actions are weighted similarly. The generative model is therefore trained to approach the original action distribution in the dataset. This is what we mean by "finding all possible actions for every state".
> > > >
> > > > During the training, we update the Q-function and the latent policy in each iteration, and therefore, we can better evaluate the advantage of every state-action pair.
> > > > While learning the Q function and the policy, discounted values will be propagated to the state-action pairs along the trajectory that connects to the high-return state-action pairs in the dataset. Actions that lead to such states will have higher Q-values and thus higher advantage weights. The generative model is then trained to learn the action distributions that are biased towards these high-return actions, and the latent policy picks the optimal action from the actions generated by the generative model.
> > > >
> > > > The task "maze2D-medium", "maze2D-medium", "antmaze-medium-diverse" and "antmaze-large-diverse" in our experiments are exactly this case, where trajectories are commanded towards different locations, and only a few actions in the dataset have rewards.
> > > > According to Table 2, LAPO is able to learn a policy in such cases that significantly outperform other methods.
> > > >
> > > >
> > > > **Q3: Why couldn't we just use "deterministic policy + advantage weighted training"?**
> > > >
> > > > Existing offline RL methods can hardly learn a successful deterministic policy with advantage weighted training in heterogeneous settings (e.g., AWAC, AWR, IQL). They either suffer from generating OOD actions between multiple modals, or fail to learn the optimal solution due to explicit constraints to the low-quality behavioral policy.
> > > >
> > > > LAPO is able to learn a deterministic policy that satisfy these two requirements.
> > > > In order to learn the deterministic policy, we iteratively train a generative model, given the weighted dataset, to provide conservativeness to the output of the policy, then evaluate, and update the latent policy by directly maximizing the Q value.
> > > > Without the data-constrained generative model, the output of the policy can be out-of-distributed, which causes errors in learning the Q function; and without the latent policy that directly optimizes the Q value, the output action can be sub-optimal.
> > > >
> > > > Please refer to the result in Table 2, showing that LAPO outperforms other methods on almost all heterogeneous tasks, including IQL, which learns a deterministic policy and applies advantage weighting.
> > > > Please note that the latent policy is a deterministic policy, condition on a state, deterministically outputs a latent value which is also deterministically mapped to the action space using the generative model.

---

> > > > > ### Author Response · Authors · 2022-08-09
> > > > > **Response to Reviewer cCVP (2)**
> > > > >
> > > > > **Q4: Why the decoder can avoid OOD actions?**
> > > > >
> > > > > In theory, the objective of VAE is to maximize the log-likelihood of data in the training dataset given latent values sampled from a prior distribution $p(z)$. Therefore, from an optimal decoder trained in VAE, it is unlikely to generate OOD samples with in-distribution $z$ inputs.
> > > > > To verify the practical performance of the VAE decoder in a heterogeneous setting, we conducted an additional experiment using the example provided by the reviewer.
> > > > >
> > > > > We train a VAE with a 2-dimensional latent space to approach the mixture distribution of two Gaussians: $0.5 N(-10,1)+0.5 N(10,1)$. We plot the histogram of the generated points by the decoder, and the input latent $z$ value in Figure 9 in Appendix.C (supplementary material). The points are generated from 5,000 $z$ values sampled from a truncated unit Gaussian limited to [-2, 2].
> > > > > We observed that there are only 19 samples out of 5,000 located between [-5, 5], which can be considered as OOD to the given training distribution. The $z$ values that generate these actions lie in a line in the latent space between the two areas associated with the two modes.
> > > > > We confirmed that, in practice, it is possible to generate OOD actions from the decoder. However, most of the generated samples are in-distributed, which is consistent with the theory.
> > > > >
> > > > > According to our experimental result on heterogeneous tasks in Table 2, we argue that the small number of potential OOD actions in the latent space has minor effects on our method.
> > > > > We have updated this experiment with the results in the Appendix.C in the supplementary material.
> > > > >
> > > > >
> > > > > **Q5: The reconstruction obtained from an optimal decoder of a VAE is a convex combination of examples in the training data.**
> > > > >
> > > > > We clarify the observation in prior works [3] and [4] that: the reconstruction of a given $z$ obtained from an optimal decoder of a VAE is a convex combination of samples in the training data, where the weight for each sample $x$ equals the density ratios of the sample being encoded as the given $z$ (i.e., $w(z,x) = q_\psi(z|x)/\sum_i^N q_\psi(z|x_i)$, where $N$ is the total number of training samples).
> > > > >
> > > > > It implies that the reconstruction of a $z$ can generalize between training samples that are encoded at similar locations as the given $z$ in the latent space. As shown in our plot of the new example in Figure 9 Appendix.C (supplementary material), the samples associated with the two modes are encoded on two sides of the latent space. As a result, except for a few samples in the middle, most reconstructions of z values do not interpolate between modes (leading to OOD), but remain in-distributed to one of the modes.
> > > > >
> > > > >
> > > > > **Q6: The mapping between $a$ and $z$ is changing. The training mechanism of the proposed method remains unclear.**
> > > > >
> > > > > The policy learning can be affected if the mapping between $a$ and $z$ changes drastically in each iteration.
> > > > > Therefore, in our implementation, we update $\theta'$ with a much slower frequency than $\theta$ to keep changes in the decoded actions small after each update.
> > > > > The learning curves in Figure 6 and Figure 7 in Appendix B.1 (supplementary material) show that the performance of most tasks improves over iterations even compared to other methods that do not use a latent policy.
> > > > > The results indicate that our proposed mechanism of iteratively training the generative model and the latent policy can stably learn the Q function and consistently improve the policy performance.

---

> > > > > > ### Comment · Reviewer_cCVP · 2022-08-09
> > > > > > **Response to the authors**
> > > > > >
> > > > > > Dear authors,
> > > > > >
> > > > > > Thanks for your reply. Unfortunately I still have some concerns.
> > > > > >
> > > > > > > Please note that the latent policy is a deterministic policy, condition on a state, deterministically outputs a latent value which is also deterministically mapped to the action space using the generative model.
> > > > > >
> > > > > > I think you are explaining LAPO here?
> > > > > > From Eq. (6) in the updated manuscript, it seem like everything is stochastic. Otherwise, how would the  log-likelihood  be calculated? Also in Line 162, you have "$\pi_v(z \mid s)$ can be modeled as a truncated Gaussian," which is also stochastic. I don't quite understand this response.
> > > > > >
> > > > > > Besides, the example of mixture of two Gaussians in my previous comment is meant to be a simple example where CVAE can produce OOD actions, which is confirmed by the authors. I think CVAE is insufficient for (more complicated) multi-modal distributions. Saying that, I appreciate the authors for adding new experimental results.
> > > > > >
> > > > > > Further, I have no doubt on the empirical results of the proposed method, but the interplay of the various components in the training process is still unclear to me.

---

> > > > > > > ### Author Response · Authors · 2022-08-10
> > > > > > > **Response to Reviewer cCVP**
> > > > > > >
> > > > > > > We consider the type of latent policy as an implementation choice, which can be implemented as deterministic or non-deterministic.
> > > > > > > We also mentioned in Line 162 that "For deterministic versions of the latent policy, the policy output can be limited to [−zmax, zmax] using a tanh function", and we implemented the deterministic version.
> > > > > > >
> > > > > > > The latent policy network takes the state as input and outputs the $z$ value directly. Also, when extracting action $a$ from the decoder $p_{\theta'}(a|s,z)$, we use the mean action instead of sampling an action from the distribution, which is also deterministic. This is what we mean by "latent policy is a deterministic policy."
> > > > > > >
> > > > > > > Regarding CVAE, we agree that in practice, it cannot 100% avoid OOD actions compared to method like IQL (Offline reinforcement learning with implicit q-learning), which only draw actions from the dataset when learning the Q function. However, all methods have their limitations when learning from heterogeneous datasets, and LAPO provides a good compromise in balancing OOD action avoidance and action maximization, yielding stable and promising results on a wide range of tasks.
> > > > > > > We leave further improvements on CVAE to future work.
> > > > > > >
> > > > > > > We hope that we have addressed your question about deterministic latent policy and we appreciate your valuable comments.

---

### Official Review · Reviewer_A7nB · 2022-07-09

**Rating:** 6
**Confidence:** 4
**Soundness:** 3 good
**Presentation:** 3 good
**Contribution:** 3 good

**Summary:**

The paper presents a method, latent-variable advantage-weighted policy optimization (LAPO), that learns a policy on top of an advantage-weighted latent space. In summary, the policy outputs a truncated Gaussian distribution over latent actions, which are decoded by a learned decoder. The key idea is that with this method, the agent can learn good policies from multi-modal data, or data collected by a number of demonstrators possibly completing different tasks. The paper also presents experiments suggesting LAPO does in fact perform well on heterogenous data compared to other offline algorithms.

**Questions:**

- Was I correct in assuming that multi-walker and walker-mix are the same environment?
- Can you clarify the purpose of the experiments mentioned in the introduction, and possibly provide more details about them?
- In order to prepare the paper for publication, I would strongly suggest addressing all of the presentation issues mentioned above.

**Limitations:**

Most of the limitations of the work are related to presentation, which the authors obviously have not addressed. The algorithm itself seems to be novel and the results suggest it could be a significant paper.

**Strengths And Weaknesses:**

The paper presents a novel, interesting method that seems to do well empirically in some difficult settings. However, there are issues with presentation that need to be addressed before the paper is ready for publication.

# Strengths

 - Originality: the paper appears to present a novel algorithm. It is most similar to [53], but learns an advantage weighted latent space that empirically performs better that [53] in the domains considered.
 - Significance: The paper considers the offline RL problem, which is highly relevant. A key issue for many offline learning algorithms is dealing with multi-modal data, and this paper seems to take a convincing step towards addressing this problem, making it significant.

# Weaknesses
 - The paper's introduction was confusing. It seemed to be referencing experiments done in a toy environment that are not mentioned elsewhere in the paper. The authors seem to be using this to motivate their method, but the end result was unfocused.
 - The methods section's organization was confusing. The discussion around Equation 3 was fairly confusing without the explanations that came in the later sections, and looking at Algorithm 1 it appears that equation wasn’t actually used anywhere, with LAPO instead alternating between optimizing equations 4, 2/5 and 6.
 - Some of the claims could be better substantiated. Although the results are compelling, it would be helpful to see experiments done with more than 3 random seeds. Also, the fact that AWAC+GMM does not work well does not satisfactorily prove to me that GMMs are not as well fit for this problem as the learned latent embedding.
 - The paper suffers from some issues with presentation quality that distract from the purpose of the work and make it, in its current state, unpublishable. Namely:
1. Figure 3 does not have any axis labels
2. Figures 5 and 6 are scaled sloppily, and the text is warped
3. There is some inconsistency in the results. Why are there different names for the walker2d task between figure 3 and table 1? For Walker2d-mix-jump-v1, why does AWAC win in the table while LAPO wins in the graph?
4. In section 3, it seems as though $Q$ is referring to $Q_\phi$ although that is not written
5. In section 3, $\lambda$ is used multiple times before being introduced.
6. The legend for figure 6 does not agree with the caption
7. The word 'tenerative' should be replaced with 'generative' on line 194

---

> ### Author Response · Authors · 2022-08-02
> **Response to Reviewer A7nB**
>
>
> Thanks for your constructive and detailed feedback. We address the comments below.
>
> **Q1:**
> The toy experiment is designed to demonstrate the problem of existing offline RL methods in heterogeneous settings.
> To better illustrate the performance gap, we added a table to the introduction listing the average returns obtained for each policy.
> The table is also presented below.
> It shows the performance of policies (or policy components) in the toy navigation task learned by AWAC, PLAS, and LAPO, averaged by $1,000$ samples. The agent receives a reward of 1.0 to go left [-1.5, -0.5], 10.0 to go right [0.5, 1.5], and 0.0 otherwise. The results showed that both AWAC and PLAS failed on this simple toy task, while LAPO successfully learns the optimal behavior using the latent policy.
>
> |                | **AWAC** |      |   | **PLAS** |        |   | **LAPO** |       |
> |----------------|----------|------|---|------|--------|---|------|---------------|
> |                | Gaussian | GMM  |   | VAE  | policy |   | VAE  | latent_policy |
> | Average return | 2.43     | 4.39 |   | 1.03 | 1.00   |   | 4.24 | 10.00         |
>
>
> **Q2:**
> Equation 3 aims to give an overall objective for the method. We have revised L108-123 (L129-142 in the revised version) of the method section to improve clarity.
>
> **Q3:**
> We have revised the statement on GMMs to "These results suggest that the VAE is more robust to model the high-reward regions in heterogeneous data."
>
> Regarding running experiments with more seeds, we have updated the LAPO results in Tables 1 and 2 with five random seeds. We will also update the results of other methods with more seeds as soon as the results are ready.
> The LAPO results with 5 random seeds are presented below:
>
> | Task ID                   | LAPO (5 seeds) |   | Task ID               | LAPO (5 seeds) |
> |---------------------------|----------------|---|-----------------------|----------------|
> | Walker2d-mix-forward-v1   | 69.66 ± 23.77  |   | Hopper-random-v2      | 19.14 ± 11.58  |
> | Walker2d-mix-backward-v1  | 102.78 ± 1.57  |   | Walker2d-random-v2    | 1.65 ± 1.41    |
> | Walker2d-mix-jump-v1      | 44.64 ± 9.4    |   | Halfcheetah-random-v2 | 30.97 ± 0.91   |
> | Maze2d-umaze-v1           | 109.5 ± 38.31  |   | Hopper-medium-v2      | 55.75 ± 4.52   |
> | Maze2d-medium-v1          | 155.62 ± 27.82 |   | Walker2d-medium-v2    | 79.81 ± 2.41   |
> | Maze2d-large-v1           | 191.55 ± 28.22 |   | Halfcheetah-medium-v2 | 46.17 ± 0.43   |
> | Antmaze-umaze-diverse-v1  | 94.0 ± 4.53    |   | Hopper-expert-v2      | 110.9 ± 2.15   |
> | Antmaze-medium-diverse-v1 | 84.0 ± 11.54   |   | Walker2d-expert-v2    | 111.89 ± 0.8   |
> | Antmaze-large-diverse-v1  | 60.0 ± 17.16   |   | Halfcheetah-expert-v2 | 95.27 ± 0.26   |
> | Kitchen-complete-v0       | 53.5 ± 8.51    |   |                       |                |
> | Kitchen-partial-v0        | 59.0 ± 5.25    |   |                       |                |
> | Kitchen-mixed-v0          | 61.5 ± 7.72    |   |                       |                |
>
>
> **Q4:**
> We have added axis labels to Figure 3 and 4, re-scaled Figure 5 and 6, fixed the typos, and fixed the task name of walker2d.
>
> In Figure 3 (Figure 6 in the revised version), the last performance of Walker2d-mix-jump-v1 using AWAC is in fact higher than LAPO. However, it is not very visible due to the small figure size. We have replaced Figure 3 with one that has better resolution and moved it to Appendix B.1.

---

> > ### Comment · Reviewer_A7nB · 2022-08-04
> > **Response from Reviewer A7nB**
> >
> > Thank you for the detailed response and for addressing my concerns. The paper looks much better, and I'll update my review accordingly.

---

### Official Review · Reviewer_eXjC · 2022-07-13

**Rating:** 6
**Confidence:** 4
**Soundness:** 3 good
**Presentation:** 3 good
**Contribution:** 3 good

**Summary:**

The authors propose LAPO, an offline-RL algorithm that is designed to efficiently handle the scenario where the provided dataset has heterogeneity. In this setup, the dataset is composed of trajectories from policies solving tasks that could be different from the target task, and/or trajectories from policies solving the target task but in behaviorally diverse ways. There are two main models in LAPO that are jointly trained – 1.) a latent-variable generative model that maximizes the weighted data log-likelihood, where the weight is computed using the advantage function; 2.) a latent policy that is learned to maximize the long-term discounted cumulative rewards (Q-value). Through experiments on the standard D4RL benchmarks (locomotion, navigation, manipulation), the authors show that LAPO compared favorably to the baseline offline RL methods, especially when the dataset is heterogeneous.

**Questions:**

1. The crucial points of differentiation between LAPO and PLAS are: a.) advantage weighting that is applied in equation 4; b.) the iterative training of the generative model. As the authors mention in related-work, the proposal of a latent-variable generative model and a latent-policy was done in PLAS. When these concepts are introduced in Section 3, I find the corresponding references to PLAS missing.
For quantitative comparisons to PLAS, since LAPO is compared with the “original” PLAS code, could there be other differences in the model implementations (decoder, Q-function, policy) that are having an impact? Is this a reasonable way to ensure parity – run the LAPO code with $\omega=1$ and move the decoder update out of the training loop, keeping everything else the same?

2. Could the authors provide some intuition as to why the latent policy can avoid OOD actions by using the VAE decoder? The approach is understandable when the action distribution is uni-modal. However, with multi-modal action distributions, could using VAEs – that tend to be mode-covering – potentially lead to sampling OOD actions?


**Limitations:**

Yes, the authors have mentioned the limitations in the checklist.

**Strengths And Weaknesses:**

The problem setting considered in this paper – offline-RL with heterogeneous datasets – seems to be under-explored in the community. The paper is well-written, well-motivated (e.g., Figure 2), and the experiments/ablations look thorough. In terms of originality, I am inclined to rate the paper as “incremental” as the proposed method looks to be a derivative of PLAS and prior advantage-weighted BC methods.

---

> ### Author Response · Authors · 2022-08-02
> **Response to Reviewer eXjC**
>
>
> Thank you for your constructive and thoughtful feedback. We address the comments below.
>
> **Q1:**
> We added the reference to PLAS in the method section (L136) and a discussion in the Appendix A.1 to describe the connection and difference to PLAS.
>
> To provide a quantitative comparison between LAPO and PLAS, we conduct an additional ablation study to show how the dynamic re-weighting mechanism is affecting the performance.
> we evaluate the learning performance on heterogeneous tasks using $\omega^+ \in$ {0.5, 0.7, 0.9, 0.99}. In our implementation, we set the importance weight to actions with positive advantages as $\omega^+$ and actions with negative advantages as $\omega^-=1-\omega^+$.
> When $\omega^+=0.5$, all actions are equally weighted, and the method becomes identical to PLAS, but shares the same network structure and data processing techniques with LAPO.
> As $\omega^+$ increases, more weights are added to the actions with positive advantage, and when $\omega^+=0.99$, the generative model tends to ignore the actions with negative advantage.
> The normalized results with a 95\%-confidence interval are presented below. The scores are averaged over the final 10 evaluations and 3 seeds.
>
> The PLAS implemented in our code structure performs better than the original code, probably because we have better hyper-parameters and did proper data normalization. Still, we observe a significant performance improvement when we increase $\omega^+$ over 0.5. This result suggests that the dynamic re-weighting mechanism plays an important role in the algorithm, making LAPO outperform PLAS when learning from heterogeneous data sets.
>
> | Task ID                   | $\omega^+$=0.5 (PLAS) | $\omega^+$=0.7            | $\omega^+$=0.9            | $\omega^+$=0.99           |
> |---------------------------|---------------------|-------------------------|-------------------------|-------------------------|
> | Walker2d-mix-forward-v1   | 42.86 ± 34.77       | 71.28 ± 16.8            | **74.17 ± 22.64**       | 61.2 ± 26.94            |
> | Walker2d-mix-backward-v1  | 85.65 ± 2.67        | 98.76 ± 1.41            | 99.22 ± 5.19            | **103.35 ± 1.31**       |
> | Walker2d-mix-jump-v1      | 2.87 ± 30.08        | 38.12 ± 4.48            | 43.2 ± 3.8              | **44.02 ± 10.79**       |
> | Maze2d-umaze-v1           | 58.3 ± 23.41        | **128.63 ± 24.49**      | 118.86 ± 55.66          | 108.2 ± 37.85           |
> | Maze2d-medium-v1          | 92.74 ± 22.85       | 123.37 ± 20.98          | 142.75 ± 11.67          | **148.11 ± 24.58**      |
> | Maze2d-large-v1           | 133.35 ± 15.32      | 181.95 ± 40.83          | **200.56 ± 18.86**      | 146.76 ± 43.82          |
> | Antmaze-umaze-diverse-v1  | 89.33 ± 9.12        | 86.67 ± 8.54            | 91.33 ± 10.67           | **94.67 ± 1.07**        |
> | Antmaze-medium-diverse-v1 | 66.67 ± 21.66       | **86.33 ± 7.0**         | 85.67 ± 9.24            | 83.33 ± 9.3             |
> | Antmaze-large-diverse-v1  | 50.33 ± 13.62       | 63.33 ± 6.49            | 61.67 ± 21.34           | **65.33 ± 19.76**       |
> | Kitchen-complete-v0       | 51.83 ± 5.94        | 52.67 ± 6.15            | **53.17 ± 9.62**        | 52.67 ± 7.07            |
> | Kitchen-partial-v0        | 46.67 ± 2.54        | 49.83 ± 3.71            | **53.67 ± 12.22**       | 51.33 ± 16.33           |
> | Kitchen-mixed-v0          | 52.13 ± 3.08        | 55.5 ± 4.55             | **62.42 ± 7.06**        | 60.33 ± 6.7             |
>
>
> **Q2:**
> The VAE is trained to maximize the log-likelihood of actions in the given dataset. Given a fully trained VAE model for which the VAE loss is minimized, the decoder should produce in-distribution samples when in-distribution states and latent values are given as inputs. The states come from the offline dataset, and the latent samples are limited to stay in distribution when querying the generative model. Please see our ablation study ("limiting the latent values" L296), in which we show the effect of sampling out-of-distribution latent variables on the policy training performance.
> Prior works on VAE training also confirm our view that VAEs models are unlikely to generate OOD samples. It is suggested in [3] (Can VAEs Generate Novel Examples? Neurips workshop 2018) and [4] (Amortized Inference Regularization NeurIPS 2018) that the reconstruction obtained from an optimal decoder of a VAE is a convex combination of examples in the training data. Therefore, the decoder is unlikely to generate OOD samples for in-distribution inputs.

---

### Official Review · Reviewer_AQNa · 2022-07-14

**Rating:** 6
**Confidence:** 4
**Soundness:** 3 good
**Presentation:** 3 good
**Contribution:** 3 good

**Summary:**

This paper investigates the heterogeneous datasets problem in offline reinforcement learning. The major idea of the proposed method LAPO can be summarized as two points. First, the authors construct the policy with a latent variable to better model multi-modal actions. Second, the advantage-weighted scheme is conducted to highlight the importance of high-reward trajectories during the training. The experimental result shows that LAPO can outperform other compared methods by a clear margin with heterogeneous datasets.

**Questions:**

See Weaknesses.

**Limitations:**

The authors have addressed the limitations of the paper.

**Strengths And Weaknesses:**

Strengths:

* The paper extends traditional offline reinforcement learning into a practical setting, where the offline datasets are heterogeneous and of diverse quality. This is good for making IL methods applicable in real-world tasks.

* The motivation is clear and the proposed method is technically sound. Moreover, the experiments demonstrate the effectiveness of the proposed method.

Weaknesses:

* The idea of using latent policy to represent multi-modal actions and advantage weighting seems to derive from existing offline RL methods.

* Some related work can be added.

  1. The method reminds me of another exponential advantage-weighted method [1] in imitation learning, which also addresses the diverse demonstrations problem. What is the difference in the advantage-weighting scheme between LAPO and [1]?

  2. [2] also addresses the problem of OOD data by weighting in offline reinforcement learning. However, [2] assigns higher weights not to the data *with high advantage* but to the data that is *close to the trained policy distribution*. Is there any contradiction between these two weighting schemes and which one is better?

  [1] Learning to Weight Imperfect Demonstrations, ICML 2021

  [2] Uncertainty Weighted Actor-Critic for Offline Reinforcement Learning, ICML 2021

---

> ### Author Response · Authors · 2022-08-02
> **Response to Reviewer AQNa**
>
> Thank you for pointing out two works that use a weighting mechanism in policy learning with the offline dataset. We have updated our related work to include the two papers. We state below the connections and differences between these two works and LAPO.
>
>
> **Paper 1**:
> Both [1] and LAPO address the challenge of learning from imperfect demonstration, and they share a similar high-level idea of "learning from data with a higher advantage."
> In [1], the imperfect demonstrations are generated by a mixture of demonstrator policies $\pi$ consisting of an optimal policy with one or several non-optimal policies (e.g., early stop checkpoints).
> A key assumption in [1] is that the advantage function $A_{\pi}(s,a)$ can be replaced by $log[\pi(a|s)]$, which suggests that actions with a higher likelihood in $\pi$ also have higher advantages in solving the target task.
> However, this assumption does not hold in the heterogeneous setting in LAPO, where $\pi$ consists of policies trained from various tasks. Actions with a high likelihood in $\pi$ are not necessarily good actions for the target task.
> Besides, [1] follows a GAIL-style training process that requires online interaction, which is not allowed in the offline setting addressed in LAPO. Regarding performance, [1] needs more than 50\% of the optimal data, while LAPO can learn from a much lower percentage.
>
>
> **Paper 2**:
> Both [2] and LAPO propose to use importance weight to address the challenge of avoiding OOD action selection in learning from the offline dataset.
> The difference is that [2] adopts uncertainty as the importance weight to reduce the contribution of OOD actions in updating the Q function and policy. It encourages the agent to learn a conservative behavior that avoids low-density actions.
> However, such conservative behavior may lead to failure on tasks with an insufficient number of optimal trajectories because the optimal actions could be down-weighted due to high uncertainty. In contrast, our method allows the agent to learn high-return actions while being conservative to the dataset.

---

### Meta-Review · Area_Chair_FcEr · 2022-09-10

**Recommendation:** Accept
**Confidence:** Certain

**Metareview:**

The paper studies the setting with heterogeneous datasets in offline reinforcement learning
The paper proposes an offline-RL algorithm that is designed to efficiently handle the scenario.

The paper received 4 expert reviews with a broad consensus on value of the problem and strengths of the method. While there is a lack of unanimous consensus, all reviewers however agree that the problem setting is of interest to broader community, and while there may be some concerns on the method, it would be to the community's benefit to see this work.

**Award:**

No

---

### Decision · Program_Chairs · 2022-09-14

Accept